

# Improving Marine Sediment Carbon Stock Estimates: The Role of Dry Bulk Density and Predictor Adjustments

Mark Chatting[1], Markus Diesing[2], William Ross Hunter[3], Anthony Grey[4], Brian P. Kelleher[4], Mark Coughlan[5,6]

[1]School of Engineering, University College Dublin, Dublin, Ireland
[2]Geological Survey of Norway, P.O. Box 6315, Torgarden, 7491 Trondheim, Norway
[3]Agri-Food and Biosciences Institute Northern Ireland, Fisheries and Aquatic Ecosystems Branch, Newforge Lane, Belfast, BT9 5PX, UK
[4]School of Chemical Science, Dublin City University, Dublin, Ireland
[5]School of Earth Sciences, University College Dublin, Dublin, Ireland
[6]SFI Research Centre for Applied Geosciences (iCRAG), O'Brien Centre for Science East, University College Dublin, Dublin Ireland

*Correspondence to:* Mark Chatting (mark.chatting@ucd.ie)

**Abstract.** Continental shelves are critical for the global carbon cycle, storing substantial amounts of organic carbon (OC) over geological timescales. Shelf sediments can also be subject to considerable anthropogenic pressures, offshore construction and bottom trawling for example, potentially releasing OC that has been sequestered into sediments. As a result, these sediments have attracted attention from policy makers regarding how their management can be leveraged to meet national emissions reductions targets. Spatial models offer solutions to identifying organic carbon storage hotspots; however, data gaps can reduce their utility for practical management decisions. Regional spatial models of OC often use global scale predictors which may have biases on regional scales. Moreover, dry bulk density (DBD), an important factor in calculating OC stock from sediment OC content, has comparatively few data points globally. We compared two spatial models of OC stock in the Irish Sea, one using unadjusted predictors and a previously used method to estimate DBD, and another incorporating bias-adjusted predictors, from *in situ* data, and a machine learning-based DBD model, to assess their relative performance. The adjusted model predicted a total OC reservoir of $46.6 \pm 43.6$ Tg within the Irish Sea, which was 31.4% lower compared to unadjusted estimates. 70.1% of the difference between adjusted and unadjusted OC stock estimates was due to the approach for estimating DBD. These findings suggest that previous models may have overestimated OC reservoirs and emphasizes the influence of accurate DBD and predictor adjustments on stock estimates. These findings highlight the need for increased in situ DBD measurements and refined modelling approaches to enhance the reliability of OC stock predictions for policy makers. This study provides a framework for refining spatial models and underscores the importance of addressing uncertainties in key parameters to better understand and manage the carbon sequestration potential of marine sediments.



## 1 Introduction

Continental shelves are important sinks of atmospheric carbon dioxide and play a key role in the global carbon
cycle (Frankignoulle and Borges, 2001). Marine sediments in these environments store substantial amounts of
organic carbon (OC) over millennia (Hage et al., 2022; Laruelle et al., 2018). Effective management of these
natural long-term stores of OC has the potential to offer policy makers a mechanism to offset emissions. As a
result, nature-based solutions to mitigating anthropogenic greenhouse gas emissions have received much scientific

interest in recent years (Griscom et al., 2017). For example, coastal vegetated habitats store >30 Pg of OC globally
and management of these habitats is thought to have the potential to offset approximately 3% of annual global
greenhouse gas emissions (Macreadie et al., 2021). Global estimates of OC stocks in continental shelf sediments
are up to nine times that of coastal vegetated habitats (between 256 to 274 Pg) (Atwood et al., 2020) and while
still heavily debated, emissions from human pressures on marine sediments are thought to be substantial (Hiddink

et al., 2023; Sala et al., 2021). Despite their large capacity to store OC, efforts to quantify stocks and potential
emissions reductions from management are relatively recent (Diesing et al., 2017; Epstein et al., 2024; Smeaton
et al., 2021). Subcontinental and national scale OC stock estimates have been undertaken, for example Diesing et
al. (2017) reported that the Northwest European continental shelf contained between 230 and 880 Tg OC stored
in the uppermost 10 cm of the sediment column and Smeaton et al. (2021) estimated that between 456 and 592

Tg of OC were stored in surficial (0 – 10 cm) marine sediments within the United Kingdom Exclusive Economic
Zone.

Despite advancements in understanding OC storage in marine sediments, data and knowledge gaps remain. One
such data gap is that of marine sediment Dry Bulk Density (DBD). DBD represents the mass of dry sediment
within a given volume of wet sediment, which is multiplied by OC content and sediment depth to calculate an

mass of OC per unit of s area, which is the OC stock (Taalab et al., 2013). DBD is a scaling factor on OC content
and adjusts the OC content in a given volume based on the density of sediment, altering OC stock estimates. Thus,
DBD has a significant effect on OC stock estimates. Previous estimates of OC stocks in terrestrial soils suggest
much of the uncertainty in overall stock estimates results from uncertainty in sediment density (Dawson and
Smith, 2007). Despite the importance of DBD in calculating OC stock, however, there remains a lack of direct

measurements for marine sediments. For example, Atwood et al. (2020) compiled a global database of ~12,000
sediment cores to predict global OC stocks and over two-thirds (69%) of their data were lacking DBD
measurements.

Subcontinental predictions of OC content are frequently based on global environmental predictors, which may
contain biases when applied to regional or smaller scales (Galmarini et al. 2019). As a result, applying bias

adjustments to model input data to align better with observational data is common practice in other scientific
disciplines, for example localised climate modelling and agricultural impact assessments (Laux et al., 2021; Luo
et al., 2018). Bias adjustments are an important component of climate modelling to reduce systematic errors in
model outputs, ensuring that projections match local conditions and are reliable for practical applications (Laux
et al., 2021). Bias adjustments have been used to improve climate model utility in agricultural impact assessments,

such as predicting planting dates and crop suitability in water-limited regions; to correct overestimations in soil
moisture models and to improve predictions in sea ice thickness (Laux et al., 2021; Lee and Im, 2015; Mu et al.,



2018). These studies collectively highlight that bias adjustments are essential for improving the precision and applicability of climate model outputs across different environmental contexts, however, their ability to adjust predictions of marine sediment OC stocks has not been investigated.

Public data repositories provide an opportunity to use data gathered over large spatial scales not practical to collect over short- and medium-term research projects (Mitchell et al., 2019). Ocean and earth sciences data, in particular, lend themselves to being collated across research groups and sampling expeditions. Much of the instrumentation and parameters measured are the same, for example temperature and salinity. In order to perform bias adjustments of globally modelled data, large datasets of parameters of interest are required (Laux et al., 2021). Public repositories, for example, the Pangaea repository of datasets (Felden et al., 2023), the International Council for the Exploration of the Seas (ICES) data centre (https://www.ices.dk/data/Pages/default.aspx) and national repositories such as Ireland's Marine Institute offer large amounts of ocean data which can be used to perform localised bias adjustments. Additionally, data specifically useful for spatial modelling of marine sedimentary OC stock, for example OC content and DBD is available from the Modern Ocean Sediment Archive and Inventory of Carbon (MOSAIC) (Paradis et al., 2023; Paradis and Eglinton, 2024).

This study aimed to determine whether bias adjusted model predictors and improved estimates of DBD could be used to improve estimates of OC stock within the Irish Sea. To address this question, the estimates of two spatial models to predict OC stock in surficial sediments in the Irish Sea were contrasted. The first model was developed using un-adjusted predictors and a widely used DBD model (Diesing et al., 2017, 2021; Smeaton et al., 2021) to estimate OC stock from OC content; and the second model was developed by bias adjusting and downscaling predictors using observational data and a machine learning spatial model of DBD (Fig. 1).

## 2 Regional setting

The Irish Sea is a shallow continental shelf sea between the land masses of the island of Ireland and Great Britain, with an average water depth of 60 m and a maximum depth of approximately 315 m (Fig. 2). The area has a complex geological history of previous glaciation coupled with marine transgression, and so the seafloor in this area consists of a mosaic of sediment types and bedforms (Arosio et al., 2023; Scourse et al., 2019; Ward et al., 2015). At present, a combination of wave and tidal current action results in a significant amount of sediment being mobilised and transported within the region (Coughlan et al., 2021). Previous studies in mapping organic carbon stocks for this region have either been coarsely resolved as part of a wider geographical study or limited to parts of the Irish Sea (Diesing et al., 2017; Smeaton et al., 2021; Wilson et al., 2018) (Crowe et al., 2023)

The study area detailed here covers a marine area of 75,229 km$^2$ and spans latitudes 50°N to 56°N and longitudes 8°W to 2°W (Fig. 2). OC content (%) (OC$_{content}$) and OC stock (OC$_{stock}$) were estimated within the study area, excluding areas within inshore waters (Smeaton et al., 2021). The inshore area excluded from the study area was defined by the Maritime Boundaries Geodatabase (Maritime Boundaries Geodatabase: Internal Waters, version 4. ).



**3 Methods**

**3.1 Organic carbon content data**

Direct measurements of sediment $OC_{content}$ were obtained from various sources, including published scientific literature, governmental organizations, as well as a private organization (Supplementary information S1). Only $OC_{content}$ data from the top 10 cm of the sediment profile were included in the analysis as the aim of the study was to estimate surficial sediment $OC_{content}$ and $OC_{stock}$. Data that reported Loss on Ignition (LOI) were converted to $OC_{content}$ using Eq. (1) (Grey et al., 2024):

$$OC_{content} = LOI \times 0.51 + 0.11, \tag{1}$$

This conversion equation was locally developed on Irish Sea $OC_{content}$ to LOI ratios where $OC_{content}$ was measured using an elemental analyser.

$OC_{content}$ data points were spatially aggregated to match the spatial resolution of the finest resolution model predictor, which was EMODNet bathymetry (approximately 155 m by 230 m cell size) later used in model training. When multiple response data points fell within a single grid cell, the mean was calculated, giving one value per grid cell.

**3.2 Predictor data**

**3.2.1 Data for bias correction**

To compare two spatial models for predicting $OC_{content}$, two predictor datasets were developed: pre-adjustment predictors ($predictors_{pre}$) and post-adjustment predictors ($predictors_{post}$) (Table 1). Potential model predictors were selected based on their anticipated relevance to $OC_{content}$ (Diesing et al., 2017, 2021). $Predictors_{pre}$ were sourced from a variety of governmental organizations and published scientific literature (Table 1). Detailed descriptions of $predictors_{pre}$ are provided in the supplementary methods.

As global scale models can have biases on regional scales (Casanueva et al., 2018, 2020; Galmarini et al., 2019; Roberts et al., 2019), $predictors_{post}$ data were developed by regionally bias adjusting and downscaling $predictors_{pre}$ data using *in situ* measurement data. Observation data from the Northwest European Shelf, rather than just the study area (Irish Sea) were used to maximize the data available for bias adjustment, resulting in regionally bias adjusted $predictors_{post}$ data. Observational data used to bias adjust $predictors_{pre}$ were sourced from public repositories: Pangaea (www.pangaea.de), The Marine Institute (https://erddap.marine.ie/erddap/tabledap/IMI_CTD.html) and MOSAIC (Paradis et al., 2023; Paradis and Eglinton, 2024) and temporally aligned with $inputs_{pre}$ data. More detail of the observational data is provided in supplementary methods.

**3.2.2 Bias adjustment**

Depending on data availability, different approaches were used to bias adjust $predictors_{pre}$. Bottom water temperature (BWT), bottom water salinity (BWS), mean and maximum bottom water velocities ($BWV_{mean}$ and $BWV_{max}$), surface chlorophyll-a, summer surface suspended particulate matter ($SPM_{summer}$) and winter surface



suspended particulate matter (SPM$_{winter}$) all followed a quantile-quantile (QQ) mapping bias adjustment approach (Casanueva et al. 2020). First, point observational data were harmonized with predictors$_{pre}$ data, which were spatially continuous averages over several years. Briefly, observation data, which represent a measurement at one point in time and space, were smoothed across time and space (Cheng et al., 2017, 2020; Cheng and Zhu, 2016).

A spatially continuous interpolated surface was then created from the smoothed data (Cheng et al., 2017, 2020; Cheng and Zhu, 2016). Original predictors$_{pre}$ data were then adjusted using the interpolated surface by QQ mapping. QQ mapping bias adjusted models have been shown to outperform un-adjusted models (Ngai et al., 2017), and are commonly used as they preserve the trends in the original model, but adjust predictions' distribution to better align with *in situ* measurements (Ngai et al., 2017). More detail of the QQ mapping approach is provided

in supplementary methods.

Since multiple models for sediment properties—mud, sand, and gravel content—exist in the study area (Mitchell et al., 2019; Stephens and Diesing, 2015; Wilson et al., 2018), they were averaged. Previous research has shown averaging multiple models can reduce error (Dormann et al., 2018). However, as sediment data are proportional, bounded by 0 and 1 and their sum must equal 1, prior to averaging mud, sand and gravel content, additive log

ratio (ALR) transformations were applied using Eq. 2 and Eq. 3 (Mitchell et al., 2019):

$$ALR_m = \log\left(\frac{mud}{gravel}\right), \tag{2}$$

$$ALR_s = \log\left(\frac{sand}{gravel}\right), \tag{3}$$

ALR$_m$ and ALR$_s$ were then averaged across the three different models (Mitchell et al., 2019; Stephens and Diesing, 2015; Wilson et al., 2018) and then back transformed to compositional data using the following Eq. 4, Eq. 5 and

Eq. 6 (Mitchell et al., 2019):

$$mud = \frac{\exp(ALR_m)}{\exp(ALR_m) + \exp(ALR_s) + 1}, \tag{4}$$

$$sand = \frac{\exp(ALR_s)}{\exp(ALR_s) + \exp(ALR_m) + 1}, \tag{5}$$

$$gravel = 1 - (mud + sand), \tag{6}$$

Mud, sand and gravel outputs above represented the final adjusted mud, sand and gravel predictors used in

predictors$_{post}$.

Adjusted current and wave orbital velocities at the sea floor were sourced directly from scientific literature as these models were locally developed using in situ measurements (Coughlan et al., 2021). Distance to coast was not adjusted as it is a simple calculation of the geographical distance for each data point to the nearest coast. Bathymetry was also not adjusted as only the EMODNet bathymetry model was used. EMODNet bathymetry

offers the highest resolution and was developed specifically for European waters (https://emodnet.ec.europa.eu/).



### 3.2.3 Validation of predictors$_{post}$

Predictors$_{post}$ were validated against observation data to assess whether the adjustment improved their agreement with *in situ* data. To avoid artificial skill, a *k* fold cross-validation approach was employed, ensuring that the validation was conducted on data not used in the bias adjustment process (Maraun and Widmann, 2018).

Specifically, the bias adjustment was performed five times, each time excluding a different non-overlapping fifth of the observation data. For each fold, the Root Mean Squared Error (RMSE) was calculated for the bias-adjusted predictor using the observation data that had been omitted from the adjustment process. RMSE represents the difference between a model's predictions and observational data and is a commonly used metric to test model performance (Milà et al., 2022). Lower RMSE values represent improvements in model performance. This was

repeated across all folds, and the mean RMSE was used to represent the overall RMSE. This overall RMSE was then compared to the RMSE of predictors$_{pre}$ to determine whether bias adjustments improve predictor accuracy (Maraun and Widmann, 2018).

### 3.2.4 Dry bulk density

DBD is the mass of dry sediment within a given volume of wet sediment and is required to calculate OC$_{stock}$ from

OC$_{content}$. While it is not a predictor for modelling OC$_{content}$, it is crucial in calculating OC$_{stock}$. Therefore, two versions of DBD were developed, one un-adjusted and one adjusted to be later combined with unadjusted and adjusted OC content predictions, respectively. Unadjusted DBD (DBD$_{pre}$) was modelled from sediment porosity using Eq. 7, Eq. 8 and Eq. 9 (Diesing et al., 2017; Smeaton et al., 2021):

$$DBD \; kg \; m^{-3} = (1 - \phi)\rho_s, \tag{7}$$

$$\rho_s = 2650 \; kg \; m^{-3}, \tag{8}$$

$$\phi = 0.3805 \; \times \; C_{mud} + 0.42071, \tag{9}$$

Where sediment porosity ($\phi$) was calculated as a function of mud content (C$_{mud}$) and assumed a grain density ($\rho_s$) of 2650 kg m$^{-3}$. By contrast, adjusted DBD (DBD$_{post}$) was spatially predicted using *in situ* data from the Northwest European Shelf and a Random Forest (Breiman, 2001) model (details in Sect. 3.3.1).

## 3.3 Model and spatial prediction

### 3.3.1 Model training

Two OC$_{content}$ models were trained to compare the effects of using pre-adjustment (OC$_{content \; pre}$) and bias-adjusted (OC$_{content \; post}$) predictors. The Random Forest (RF) algorithm was used as it has been shown to perform well for geospatial modelling (Diesing et al., 2021; Hengl et al., 2015; Meyer et al., 2018). The RF model was trained

using the Forward Feature Selection (FFS) algorithm to omit unimportant predictors (Meyer et al., 2018). FFS trains multiple RF's using all possible 2-predictor combinations. The best of these 2-predictor models is kept, and all possible 3-predictor models are trained using the already selected two predictors. The number of predictor variables is increased iteratively. Model performance is tested for each additional predictor and the process stops when none of the remaining variables decreases the model RMSE (Meyer et al., 2018). After model training,

partial dependence plots were used to visually inspect the associations between the response data (OC$_{content}$) and



the predictors deemed to be important by FFS. Additionally, DBD$_{post}$ was spatially modelled to later calculate OC$_{stock}$ from OC$_{content}$. Predictors$_{post}$ were used to train DBD$_{post}$. Similarly to OC$_{content}$, an RF was trained, which was implemented using the FFS algorithm.

### 3.3.2 Model validation

All FFS RF models (OC$_{content\ pre}$, OC$_{content\ post}$ and DBD$_{post}$) were validated using the Nearest Neighbour Distance Matching (NNDM) Leave-One-Out (LOO) Cross Validation (CV) approach (Milà et al., 2022). NNDM LOO CV matches the distance distribution functions of training to testing data to the distance distribution function of prediction to training data (Supplementary information S1 and S2). This validation approach has been shown to produce more reliable estimates of spatial model performance than random $k$ fold CV (Milà et al., 2022). Random

$k$ fold CV randomly creates train-test splits for model training testing validation, which ignores autocorrelation in spatial data and carries the high probability data points that are spatially autocorrelated may be used to train and test model performance simultaneously. Thus, such an approach is increasing the tendency for model performance to be overestimated (Milà et al., 2022). Conversely, NNDM ensures that CV is performed on data that are spatially independent of training data. In addition to NNDM LOO CV, the DBD$_{post}$ RMSE against observational data was

calculated to determine whether RF modelling to spatially predict DBD (DBD$_{post}$) was an improvement compared to modelling DBD from porosity (DBD$_{pre}$) (details in Sect. 3.2.3).

### 3.3.3 Model uncertainty

Model uncertainty was calculated for each of the OC$_{content}$ models as well as DBD$_{post}$. Uncertainty was estimated by calculating the standard deviation between 25 Random Forest (RF) predictions (Diesing et al., 2021). Response

data were divided into 25 folds, each with a 70% to 30% train/test split, resulting in 25 models. For each pixel, the standard deviation of the 25 predictions was computed. The total uncertainty was then determined by summing these standard deviations (Diesing et al., 2021).

### 3.4 Calculation of organic carbon stock and total reservoir

The spatial variation in OC$_{stock}$, which is the mass of OC stored in sediment per unit of area to a specific depth,

across the study area was calculated for each set of un-adjusted inputs (OC$_{content\ pre}$ and DBD$_{pre}$) and adjusted inputs (OC$_{content\ post}$ and DBD$_{post}$) using the following equations (Diesing et al., 2017):

$$OC\ stock_{pre}\ kg/m^2 = OC_{content\ pre} \times DBD_{pre} \times cell\ area \times depth \text{ (Eq. 10)}$$

$$OC\ stock_{post}\ kg/m^2 = OC_{content\ post} \times DBD_{post} \times cell\ area \times depth \text{ (Eq. 11)}$$

OC$_{content}$ and DBD were the predicted values from the final selected OC$_{content}$ and DBD models, respectively. Cell

area was calculated using the *cellSize()* function in the terra package (Hijmans, 2025) in R. The *cellSize()* function calculates the area covered by grid cell in the study area, rather than assuming a constant grid cell size across the study area. Depth was assumed to be a constant 0.1 m to estimate surficial sediment OC stock. This equation was applied to each grid cell across the study area.



Additionally, the total mass of OC to a specific depth within in the entire study area, termed OC reservoir, was calculated by summing OC stock (calculated above) for all grid cells in the study area. In order to parse the relative importance of $OC_{content}$ and DBD estimates to the overall $OC_{stock}$ estimate, all possible combinations of bias adjusted and non-bias adjusted OC content and DBD models were calculated.

## 4 Results

### 4.1 Data collation

#### 4.1.1 Data sourced

A total of 1670 in situ measurements of surficial sediment $OC_{content}$ were obtained from various sources within the study area (Supplementary information S3). After spatial aggregation of $OC_{content}$ data and removing data points within the excluded inshore area, 450 data points were available for model training. Observation data availability for model predictors varied significantly (Table 1). BWT had more than 300 times the amount of data as SPM, the predictor with the lowest amount of legacy data available. DBD had 642 data points across the entire Northwest

European Shelf.

#### 4.1.2 Predictor improvement: predictors$_{pre}$ vs predictors$_{post}$

Except for SPM$_{summer}$ and BWT, all predictors$_{post}$ data showed improved consistency with observation data according to RMSE comparisons (Table 1). Therefore, unadjusted SPM$_{summer}$ and BWT were used in the predictors$_{post}$ dataset for model training. The extent to which predictors$_{pre}$ were adjusted varied (Fig. 3). Mean

adjusted BWS, for example, showed little change in RMSE between predictors$_{pre}$ and predictors$_{post}$ (Table 1). Mean change in BWS was 0.09 psu compared to predictors$_{pre}$. However, SPM$_{winter}$ was adjusted to a greater degree. Mean change in SPM$_{winter}$ was -9.97 mg l$^{-1}$, which is also reflected in a greater shift in SPM$_{winter}$'s data distribution (Fig. 3). Sediment properties, mud, sand and gravel content were not changed to a large degree (Fig. 3). Mean

adjusted change from predictors$_{pre}$ to predictors$_{post}$ in $C_{mud}$, $C_{sand}$ and $C_{gravel}$ were -0.03, 0.07 and -0.04, respectively.

### 4.2 Random forest modelling

#### 4.2.1 OC$_{content}$ and DBD$_{post}$ Variable selection

Different predictors were selected during the $OC_{content}$ model training process. FFS chose five important predictors for both $OC_{content\ pre}$ and $OC_{content\ post}$ (Fig. 4). Selected predictors for $OC_{content\ post}$ were $C_{mud}$, WOV$_{max}$, chlorophyll-

a, bathymetry and distance to coast, of which, $C_{mud}$ and WOV$_{max}$ were the most important. $OC_{content\ post}$'s Mean Squared Error (MSE) increased by 52.3% and 27.9% when $C_{mud}$ and WOV$_{max}$ were respectively removed from the model (Supplementary information S4). Partial plots also showed $C_{mud}$ had a positive relationship with $OC_{content}$, while WOV$_{max}$ was inversely related to $OC_{content}$ (Fig. 4). In contrast, predictors selected for $OC_{content\ pre}$ were SPM$_{summer}$, salinity, chlorophyl-a, WOV$_{max}$ and $C_{gravel}$ (Fig. 4). SPM$_{summer}$ was the most important predictor

for $OC_{content\ pre}$, which accounted for a 62.9% increase in the model MSE when removed (Supplementary information S4).

Six important predictors were selected by RF FFS for DBD$_{post}$ (Fig. 4). Important predictors were $C_{mud}$, SPM$_{summer}$, SPM$_{winter}$, $C_{gravel}$, WOV$_{max}$ and WOV$_{mean}$. $C_{mud}$, which was inversely related to DBD (Fig. 4) was the most



important predictor, resulting in an increase in model RMSE by 44.3% when removed (Supplementary
information S4).

### 4.2.2 Model performance and predictions

$OC_{content\ post}$ ($R^2$=0.61, RMSE=0.31%) showed a slight increase in performance compared to $OC_{content\ pre}$ (Table 2,
$OC_{content\ post}$ $\Delta R^2$ = +0.03 vs. $OC_{content\ pre}$; $OC_{content\ post}$ $\Delta RMSE$ = -0.01% vs. $OC_{content\ pre}$). This similarity in
performance was reflected in comparable $OC_{content}$ predictions across the study area. Mean $OC_{content\ post}$ prediction
was $0.57 \pm 0.58$ %, whereas $OC_{content\ pre}$ was $0.65 \pm 0.65$ (Table 2). However, patterns for $OC_{content}$ predictions were
not consistently lower for $OC_{content\ post}$ (Fig. 5). For example, $OC_{content\ post}$ was predicted to be higher in areas near
the Irish coast and southeast of the Isle of Man (Fig. 5).

Importantly for calculating OC stocks, $DBD_{post}$ had a better agreement with *in situ* data compared to $DBD_{pre}$ (Table
1). $DBD_{post}$ explained 43% of the variance in the DBD point data across the NW European shelf and had an RMSE
of 187 kg m$^{-3}$. Within the study area, $DBD_{post}$ largely showed a reduction in DBD across the study area with a
mean reduction of 310 kg m$^{-3}$. In areas of known high mud content such as 'The Smalls' and the 'Mudbelt', mean
reductions in $DBD_{post}$ were even greater (506 kg m$^{-3}$) (Fig. 6).

A substantial difference in predicted total $OC_{stock}$ across the study area was found between the two trained models
(Table 2). Based on $OC_{stock\ post}$ the total OC reservoir was $46.6 \pm 43.6$ Tg in the study area, which was 68.6% (total
$OC_{stock}$ = $67.9 \pm 63.0$ Tg) of the OC reservoir based on $OC_{stock\ pre}$ (Table 2). Both adjusted and unadjusted
predictions captured similar spatial patterns in $OC_{content}$ and $OC_{stock}$ (Fig. 5 and 7). Both 'The Western Irish Sea
Mudbelt' and 'The Smalls' had comparatively high $OC_{content}$ and $OC_{stock}$ (Fig. 5 and 7). Generally, lower $OC_{content}$
and $OC_{stock}$ were predicted in deeper central parts of the Irish Sea (Fig. 5 and 7). Improvements to DBD rather
than $OC_{content}$ were shown to have a greater influence on total OC reservoir estimate. Combining $OC_{content\ post}$ with
$DBD_{pre}$ reduced the total OC stock estimate by 6.5 Tg, whereas, combining the $OC_{content\ pre}$ with $DBD_{post}$ reduced
the total OC reservoir estimate by 15.1 Tg across the study area.

### 5 Discussion

Our findings show that bias-adjusted model inputs substantially reduced estimates of organic carbon (OC) stock
in surficial sediments within the Irish Sea by almost one-third (31.4%). Adjusted inputs showed better alignment
with *in situ* measurements and predictions for $OC_{content\ post}$ and $DBD_{post}$ had lower error compared to predictions
using non-adjusted inputs. Our results show that RF modelling of DBD data, instead of modelling DBD as a
function of porosity, led to the greatest reductions in OC stock estimates. These findings suggest that previous
wider-scale modelling efforts of OC stock, which modelled DBD from porosity, might have overestimated OC
stock. Moreover, these findings highlight the need to reduce uncertainties in model inputs to improve predictions
and make model outputs more robust to support policy makers and marine planning decisions. Our study
contributes to the refinement of spatial models for predicting marine sediment OC stocks by using improved
predictors and inputs.

Approximately two-thirds (70.1%) of the difference in OC stock estimates between the two estimates ($OC_{stock\ pre}$
vs. $OC_{stock\ post}$) was attributed to adjustments in DBD and the remaining difference was due to adjustments in OC



content model predictions. DBD$_{post}$ showed reduced error, compared to DBD$_{pre}$ and revealed consistently lower DBD values across the study area, resulting in lower OC stock estimates (DBD$_{post}$ mean 1191 ± 175 kg m$^{-3}$; DBD$_{pre}$ mean: 1501 ± 65kg m$^{-3}$). Apart from recent work which used a machine learning model to estimate DBD (Diesing et al., 2024), previous work has largely focused on accurately modelling OC content estimates, with less attention being given to DBD estimates (Diesing et al., 2017, 2021; Smeaton et al., 2021). For example, previous

work has modelled DBD from porosity as was performed in the OC$_{stock\ pre}$ model developed for the current study. Modelling DBD in this way does not utilize *in situ* measurements of DBD and reductions in DBD$_{post}$ RMSE compared to DBD$_{pre}$ in the current study suggests that modelled DBD from porosity may also be less accurate than RF modelling. Additionally, Atwood et al. (2020) used a transfer function to estimate DBD from OC content, however, the transfer function was not based solely on marine sediment data and contained OC content values

substantially greater than those observed on continental shelves. Previous research has shown that OC storage dynamics varies from inland to coastal to shelf sediments (Smeaton et al. 2021). Our findings suggest that modelling DBD from porosity may tend to overestimate DBD estimates, especially in high mud content areas. These findings highlight the importance of reducing uncertainties around DBD and reinforces prior suggestions for standardized measurement protocols, particularly regarding DBD, which influences OC stock estimates

(Graves et al., 2022).

    Previous research has consistently highlighted mud (the sum of silt and clay) content (C$_{mud}$), as a critical predictor of OC content (Diesing et al., 2017; Smeaton et al., 2021). In agreement with previous work, OC$_{content\ post}$ indicated that C$_{mud}$ was the most important predictor of OC$_{content}$. Muds across fjords and coastal sediments have been shown to contain greater amounts of OC than sand, coarse and mixed sediments (Smeaton et al., 2021). The clay fraction

in marine muds offers a large surface area for the adsorption and preservation of organic matter, making it a key factor in OC sequestration (Babakhani et al., 2025; Keil and Hedges, 1993). The capacity for sediments to bind OC through clay-OC interactions can also vary with different mineral phases occurring in sediments, varying in the surface charge and distribution, topography and particle size and subsequent geochemical conditions constraining these characteristics (e.g. pH and ionic strength of pore water) (Kleber et al., 2021). (Bruni et al.,

2022)(Hunt et al., 2020; Smeaton and Austin, 2019)

    Despite our dataset showing a largely positive relationship between C$_{mud}$ and OC$_{content}$, extremely low C$_{mud}$ values (<0.05% C$_{mud}$) were also associated with high OC$_{content}$, which is in contrast to previous work reporting positive relationship between C$_{mud}$ and OC$_{content}$ (Diesing et al., 2017; Smeaton et al., 2021). In continental shelves relationships between mud and OC$_{content}$ are complex. Previous work has shown little variation in OC$_{content}$ between

mud, sand and coarse sediments on shelf areas (Smeaton et al., 2021). However, the lability of organic matter can vary significantly between these environments (Smeaton and Austin, 2022). Marine muds have been shown to store organic matter ranging from highly reactive to highly resistant to degradation, whilst coarser sediments have been shown to almost exclusively house organic matter highly resistant to degradation (Smeaton and Austin, 2022). Furthermore, muddy sediments tend to be sites of relatively high infaunal biomass and these benthic fauna

in combination with microbial metabolism play a key role in mediating OC mineralisation and preservation (Lin et al., 2022). For example, Zhang et al. (2024)estimated bioturbation-induced remineralisation to account for between 25 and 30 % of total seabed respiration. These biological processes act alongside sediment disturbance from commercial fishing to create this nuanced relationship between mud and organic matter content (Epstein and



Roberts, 2022; Zhang et al., 2024). This may explain why the mud partial plot did not exhibit a clear positive
relationship, as the heterogeneity in organic matter lability can affect OC storage capacity.

In addition, the importance of $WOV_{max}$ in our model highlights the role of hydrodynamic conditions in shaping
OC content and stocks. The inverse relationship between OC content and $WOV_{max}$ found by the current study is
in agreement with previous work that demonstrated lower OC accumulation rates are associated with
environments with increased hydrodynamic activity (Song et al., 2022). These regions, characterized by thick
Sediment Mixed Layers (SML), experience more frequent sediment resuspension, which limits OC accumulation.
These mixing regimes facilitate the repeated suspension of fine sediment particles with varying densities and
exposure of associated organic matter to oxygen, potentially increasing remineralization and reducing organic
carbon accumulation rates (Song et al., 2022). Several knowledge gaps remain regarding the processes governing
carbon mineralization in marine sediments, particularly in dynamic coastal regions. First, the mechanistic
interplay between sediment resuspension, microbial community activity, and carbon mineralization pathways
remains poorly constrained (LaRowe et al., 2020). While oxygen exposure time is a key driver of OC degradation
(Hartnett et al., 1998), the extent to which short-term disturbance events (e.g. storms or trawling) alter oxygen
penetration depth and thus carbon remineralization rates need further investigation (Bartl et al., 2025; Glud, 2008).
Additionally, the interaction between bioturbation – a critical process mixing particulate organic matter – and
resuspension driven transport of sediments across spatial scales is not well quantified in models predicting carbon
storage (Cozzoli et al., 2019). The hydrodynamic regime has a strong influence over sediment type, as high energy
environments prevent mud deposition or resuspend finer particles, while low energy environments allow fine
sediments to settle and accumulate, which is conducive to mud deposition and OC accumulation (Hanebuth et al.,
2015). Similar findings were reported by Diesing et al. (2017), where low hydrodynamic activity was positively
correlated with OC content. These insights, coupled with the present work, underscore the need to incorporate
sediment dynamics, such as sediment mixing or disturbance, into models predicting OC stock, particularly in light
of human activities such as trawling and offshore development (Epstein and Roberts, 2022)

Diesing et al. (2017), Smeaton et al. (2021) and Atwood et al. (2020) all reported better model accuracy compared
to those in the present study. For example, Diesing et al. (2017) and Atwood et al. (2020) reported $R^2$ values of
75% and 76%, respectively. Despite $OC_{content\ post}$ showing improved performance compared to $OC_{content\ pre}$ and OC
stock input data (predictors and DBD) showed reduced error, model performance reported here is lower when
compared to previous studies investigating $OC_{stock}$ in marine sediments. These apparent differences in model
performance may be due to the validation approach used and spatial autocorrelation, which may be inflating model
metrics (Milà et al., 2022). For example, the present study used the kNNDM algorithm to ensure spatial
independence between cross validation training folds, which ensures that for each train/test fold, data that are
tested on are spatially independent of test data. However, random k fold cross validation, as used by Atwood et
al. (2020) and Diesing et al. (2017), are likely to train and test on data that are spatially dependant, and thus
artificially increasing the likelihood of the model predicting correctly (Milà et al., 2022). Similarly, Smeaton et
al. (2021) who did use a form of spatial cross validation reported comparable model performance to our study
($R^2$=53%, RMSE=1.72). Smeaton et al. (2021) used 'spatial blocks' to determine train/test splits. However, these
spatial blocks were defined as ICES statistical grids, which do not ensure spatial independence between train/test
folds, unlike the kNNDM algorithm used in the present study.



Reducing model error through adjusting model input data, predictions presented here still carry uncertainty. Even though prediction uncertainty estimates were performed, there is still more uncertainty that could not be quantified. The data that was sourced was not all recorded uniformly, and some components were unavailable. For example, uncertainty in OC content data was not reported, thus we were unable to propagate those uncertainties into final OC content and OC stock uncertainty predictions. This was also true for predictor data. Thus, uncertainties in measured OC content and predictor data could not be included in final model uncertainty estimates. In addition, DBD data were lacking across the study area and only 3% (18 of 642) of all DBD

observational data used in bias adjustment were located within the study area. However, despite this, DBD estimates presented here have reduced error when compared to observational data across the Northwest European shelf when compared to estimates from porosity. Findings from the present study show spatial models of organic carbon can still be significantly improved from increased *in situ* data. Additionally, incorporating these datasets into public repositories can improve efforts to estimate organic carbon stocks by providing ground truthed data on

which to base numerical models.

## 6 Conclusion

Overall, our findings suggest that marine sedimentary OC stocks could be lower than previously estimated, a conclusion with important implications for seabed management. The findings suggest that adjusting model inputs based on in situ data, may help reduce uncertainties in model predictions. We highlight the critical role that

accurate DBD estimates play in determining OC stock. Moving forward, more comprehensive *in situ* DBD measurements and refined DBD models are essential for improving the accuracy of OC stock predictions. Alternatively, OC stocks could be calculated directly per sediment core, reducing the number of models needed to estimate OC stocks, thus reducing uncertainty in final estimates. These efforts will be instrumental in developing better strategies for managing marine sedimentary OC stocks.

**Code/Data availability**

Spatially modelled organic carbon content, stock data, and their associated uncertainties are available as a Zenodo repository (https://doi.org/10.5281/zenodo.14859982). Additionally, the bias adjusted predictor data layers developed and the random forest dry bulk density model can be accessed from Zenodo (https://doi.org/10.5281/zenodo.14859982). The underlying code used to develop these data layers and produce

spatial predictions of organic carbon content and stock is available from the "Bias-Adjusted Predictors and Random Forest Models for Organic Carbon Stock Estimation" github repository (https://github.com/markchatting/Bias-Adjusted-OC-Stock-Model.git).

**Author contributions**

**MC**: conceptualization, data curation, formal analysis, investigation, methodology, software, validation,

visualization, writing – original draft preparation, writing – review & editing. **MD**: conceptualization, data curation, formal analysis, funding acquisition, methodology, supervision, writing – original draft preparation, writing – review & editing. **WRH**: data curation, investigation, writing – review & editing. **AG**: data curation, funding acquisition, investigation, writing – review & editing. **BK**: funding acquisition, project administration,



resources, supervision, writing – review & editing. **MCo**: conceptualization, funding acquisition, investigation, methodology, project administration, resources, supervision, writing – original draft preparation, writing – review & editing.

**Competing interests**

The authors declare that they have no conflict of interest.

**Acknowledgements**

The authors would like to thank the creators and maintainers of public data repositories, specifically the ones used in this study: PANGAEA, the Marine Institute (Eoghan Daly), Modern Ocean Sedimentary Inventory and Archive of Carbon (Tessa van der Voort, Hannah Gies and Sarah Paradis) and Natural Resources Wales.

**Financial support**

This work was conducted under the QUEST project, which is carried out with the support of the Marine Institute
and the Environment Protection Agency, funded by the Irish Government (Ref: PBA/CC/21/01).

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



**Tables and Figures**

Table 1: Summary of organic carbon content and stock model inputs. Directly sourced adjustments were when the adjusted data was soured directly from literature that developed a model based on locally measured observational data. SPM data points were for all months to create monthly interpolated surfaces then they were merged to create seasonal interpolated surfaces. ΔRMSE represents the change in RMSE after QQ mapping. Negative RMSE values represent reduced error, while positive RMSE values show increased error.

| Predictor | Unit | Abbreviation | Pre adjustment source | NWE shelf data points available | Adjustment method | ΔRMSE after adjustment |
|---|---|---|---|---|---|---|
| Distance to coast | km | - | Calculated from data points | - | None | - |
| Bathymetry | m | - | EMODNet | - | None | - |
| Bottom water salinity | - | BWS | Copernicus marine data portal | 57,965 | QQ mapping | -0.01 |
| Bottom water temperature | °C | BWT | Copernicus marine data portal | 173,607 | QQ mapping | 0.00 |
| Mean bottom water velocity | m s$^{-1}$ | BWV$_{mean}$ | Copernicus marine data portal | - | Averaging | - |
| Maximum bottom water velocity | m s$^{-1}$ | BWV$_{max}$ | Copernicus marine data portal | - | Averaging | - |
| Surface chlorophyll-a | µg l$^{-1}$ | - | Copernicus marine data portal | 21,108 | QQ mapping | -1.13 |
| Summer surface Suspended Particulate Matter | mg l$^{-1}$ | SPM$_{summer}$ | Copernicus marine data portal | 542* | QQ mapping | +2.31 |
| Winter surface Suspended Particulate Matter | mg l$^{-1}$ | SPM$_{winter}$ | Copernicus marine data portal | 542* | QQ mapping | -0.85 |
| Mud content | % | C$_{mud}$ | Mitchell et al. (2019) | - | Averaging | -0.03 |
| Sand content | % | C$_{sand}$ | Mitchell et al. (2019) | - | Averaging | -0.05 |
| Gravel content | % | C$_{gravel}$ | Mitchell et al. (2019) | - | Averaging | -0.03 |
| Mean wave orbital velocity at seafloor | m s$^{-1}$ | WOV$_{mean}$ | Wilson et al. (2018) | - | Directly sourced | - |
| Maximum wave orbital velocity at seafloor | m s$^{-1}$ | WOV$_{max}$ | Wilson et al. (2018) | - | Directly sourced | - |
| Dry bulk density | kg m$^{-3}$ | DBD | Modelled from modelled porosity | 706 | Random forest modelling | -194.73 |

Table 2: Summary of outputs from models trained on non-bias adjusted data (predictors$_{pre}$) and bias adjusted data (predictors$_{post}$). Mean OC$_{content}$ represents the mean prediction value across the study area; total reservoir estimate is the total OC stock reservoir for the study area; mean DBD is the mean DBD predicted across the study area. R$^2$ and RMSE (Root Mean Squared Error) represent performance metrics used in model selection process.

| Input data | Mean DBD (kg m$^{-3}$) ± sd | Mean OC$_{content}$ (%) ± sd | Total reservoir OC estimate (Tg) ± total uncertainty |
|---|---|---|---|
| **Predictors$_{pre}$** | 1501.60 ± 66 | 0.65 ± 0.62 | 67.9 ± 62.9 |
| **Predictors$_{post}$** | 1191 ± 175 | 0.57 ± 0.58 | 46.6 ± 43.6 |



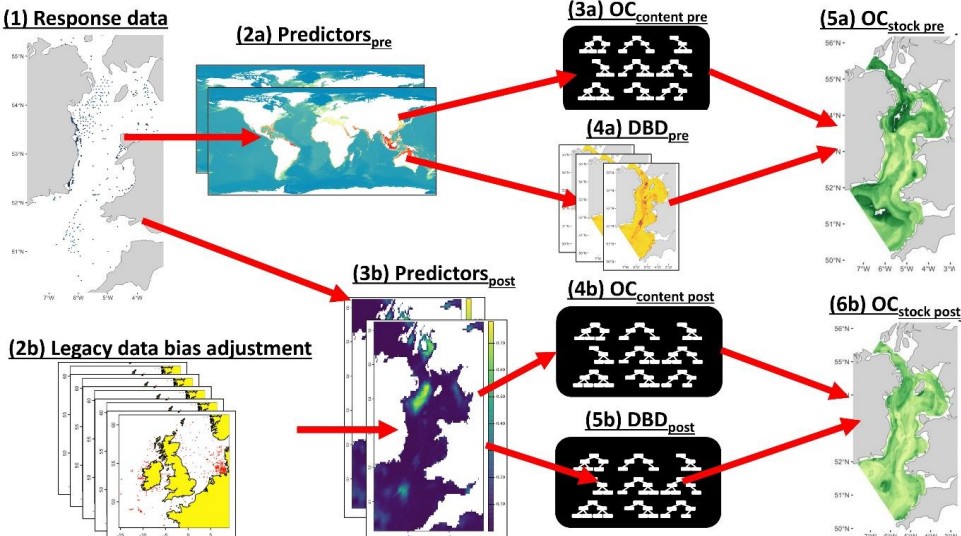

Figure 1: Summary of steps taken to train and predict form two different models, which include: 1) collating response data; 2a) compiling OC content predictor data (predictors$_{pre}$); 3a) training a random forest model to predict OC content on the non-adjusted predictor data (OC$_{pre}$); 4a) modelling Dry Bulk Density (DBD) from porosity (DBD$_{pre}$); 5a) predicting OC stock across the study area using OC$_{pre}$ and DBD$_{pre}$; 2b) bias adjusting predictors$_{pre}$ data using quantile-quantile mapping; 3b) compiling OC content predictor data after it has been bias adjusted (OC$_{content\ post}$); 4b) training a random forest model to predict OC content on the bias adjusted predictor data (predictors$_{post}$); 5b) training a random forest model to predict DBD on the bias adjusted predictor data (DBD$_{post}$); 6) predicting OC stock across the study area using OC$_{post}$ and DBD$_{post}$.



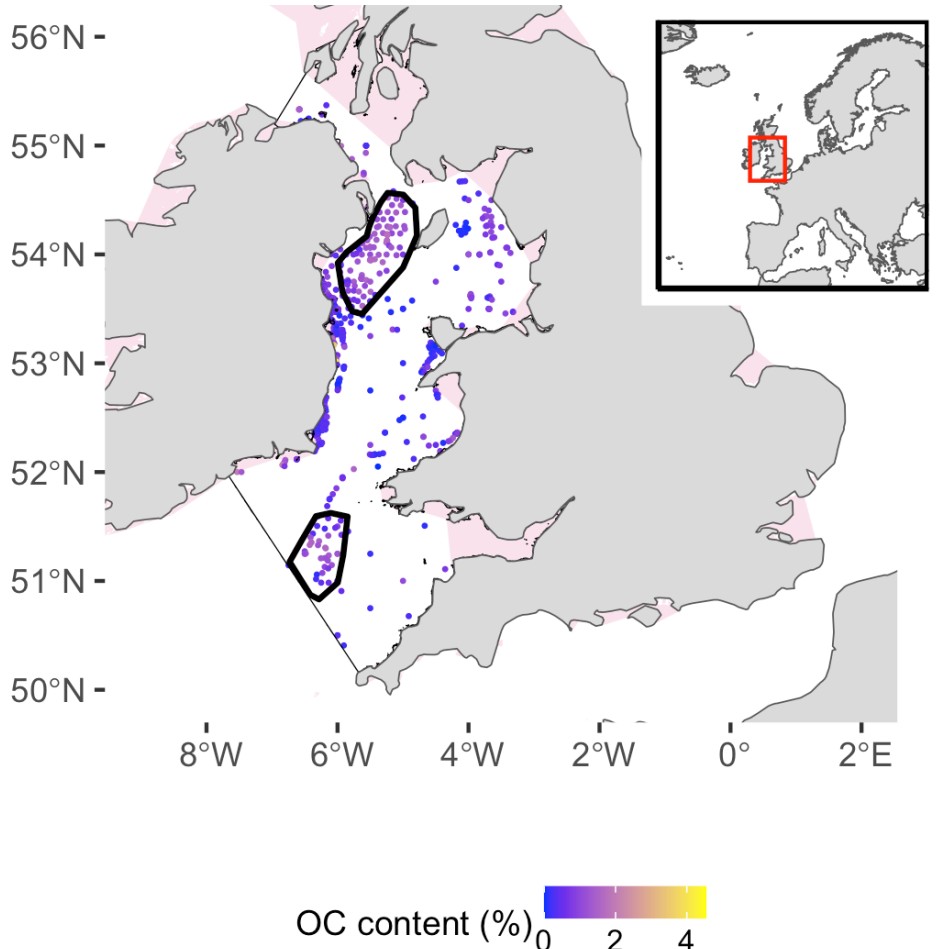

Figure 2: Study area within the Irish Sea (thin black border) and within the greater North West European shelf (inset). Points indicate organic carbon (OC) data coloured by the organic carbon content. Pink areas show internal waters that have been excluded from the study area. Thick black outlined polygons indicate the 'Mudbelt' (northern) and the 'Smalls' (southern), areas of known high mud content within the Irish Sea.



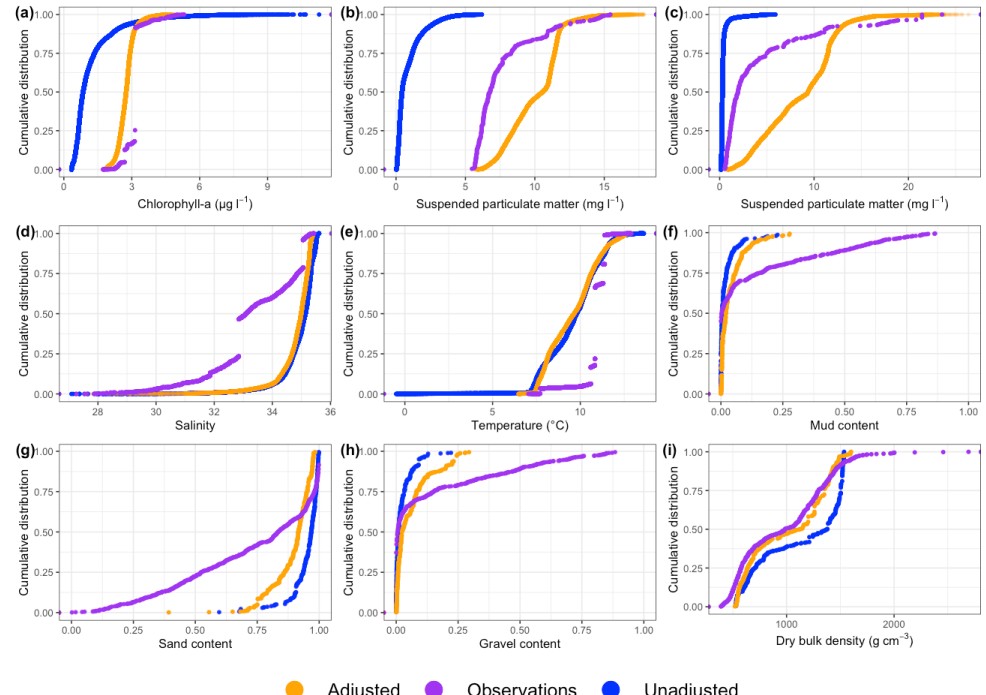

Figure 3: Cumulative distribution functions (CDF) of bias adjusted (adjusted) and not bias adjusted (modelled) model input data and observational data used in bias adjustment.



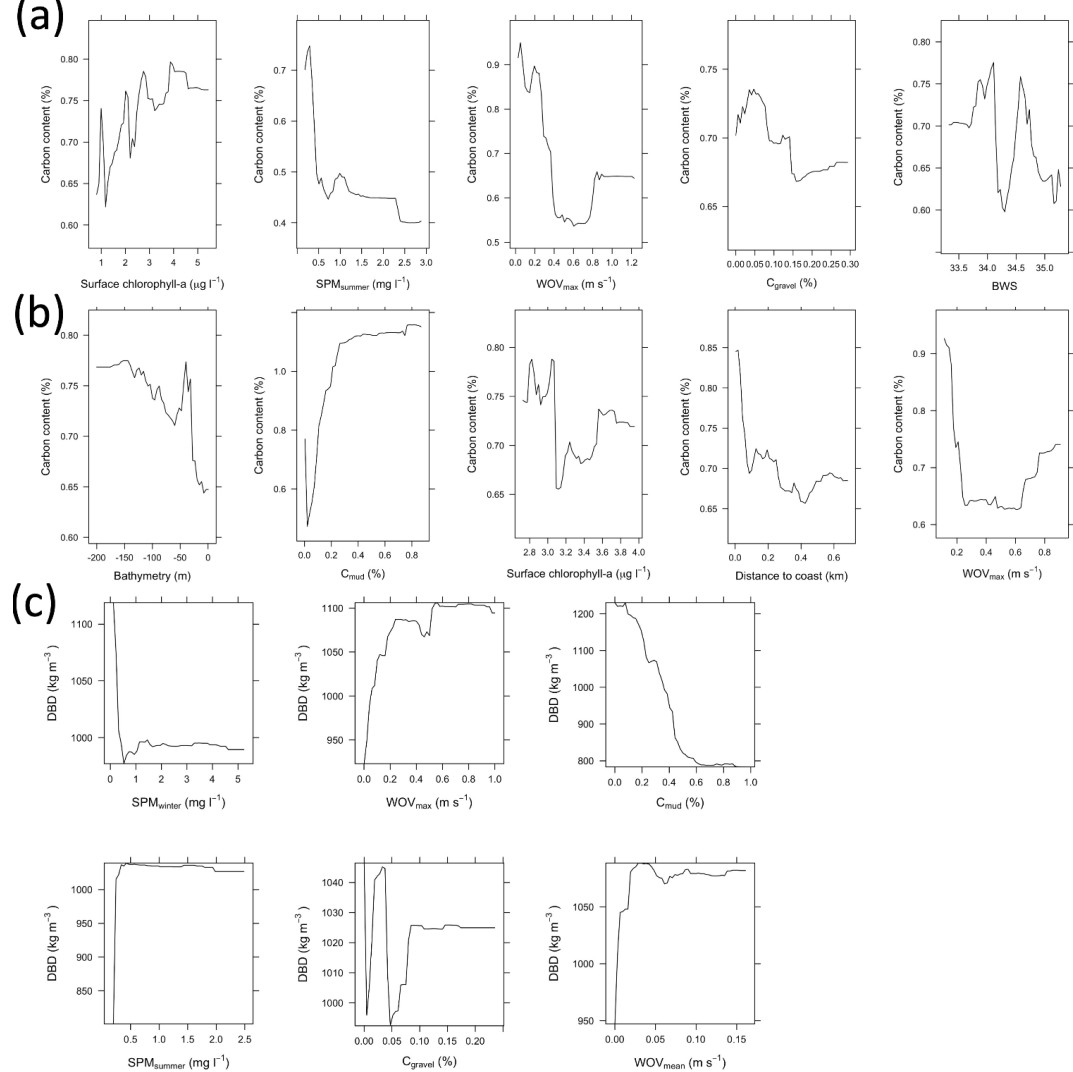

Figure 4: Partial dependence plots showing the relationship between a) OC content and non-bias adjusted model predictors selected by Forward Feature Selection (FFS): surface chlorophyll-a, surface summer suspended particulate matter, maximum wave orbital velocity at the seafloor; gravel content and bottom water salinity; b) bias adjusted predictors selected by FFS: bathymetry, mud content, surface chlorophyll-a, distance to the nearest coast and maximum wave orbital velocity at the seafloor and; c) bias adjusted predictors and dry bulk density (DBD) selected by FFS: surface winter suspended particulate matter, maximum wave orbital velocity at the seafloor, surface summer suspended particulate matter, mud content, surface chlorophyll-a and distance to the nearest coast.




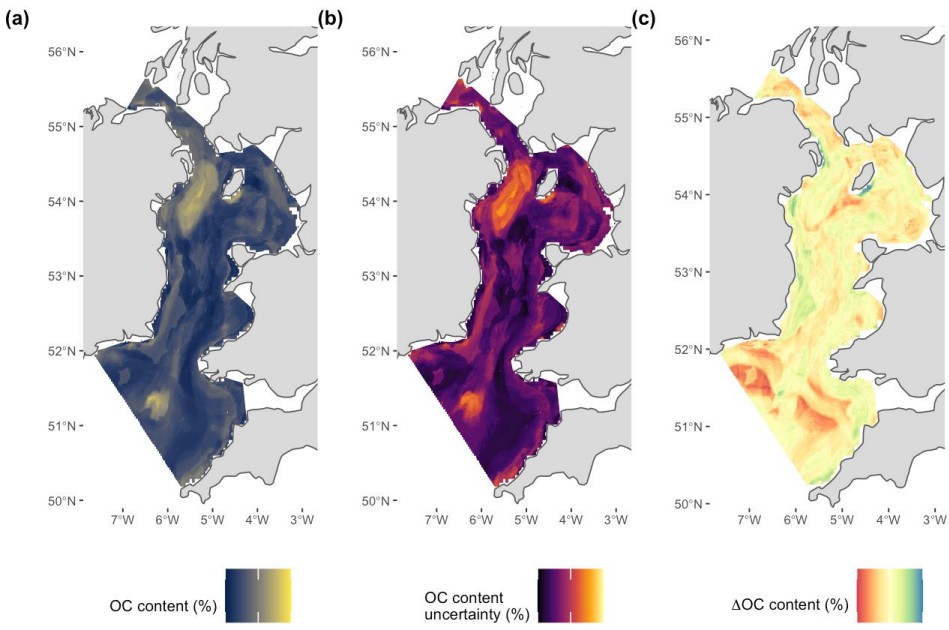

Figure 5: a) Predicted organic carbon (OC) content using adjusted model inputs; b) the associated uncertainty and c) difference between not bias adjusted and bias adjusted predictions across the study area (difference = $OC_{content\ pre} - OC_{content\ post}$). Negative values indicate where predictions with adjusted model inputs were higher than non-bias adjusted inputs.





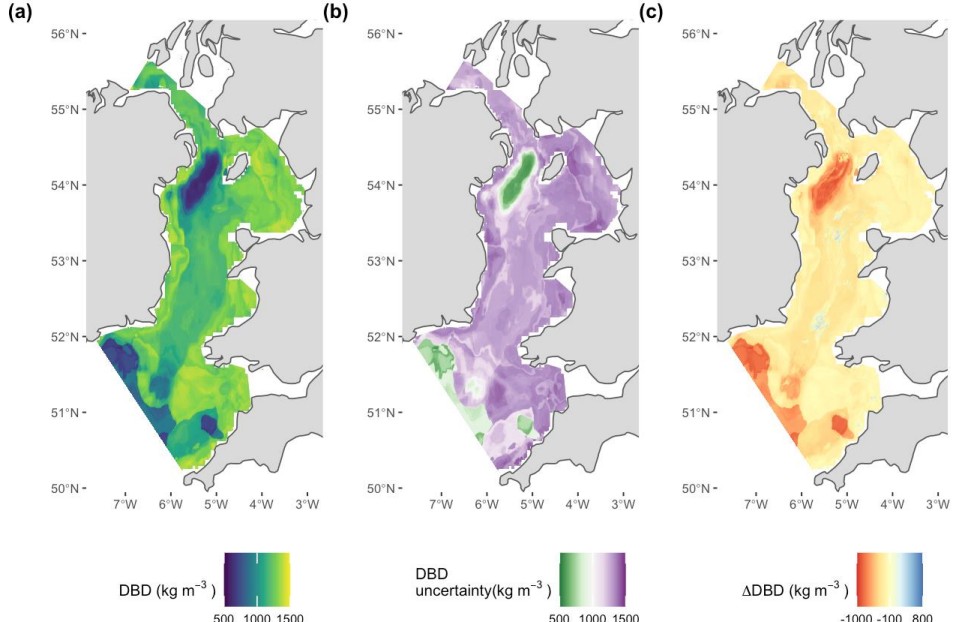

Figure 6: a) Predicted dry bulk density (DBD) content using adjusted model inputs; b) the associated uncertainty and c) difference between DBD modelled from porosity and using an RF (DBD$_{pre}$ - DBD$_{post}$). Negative values indicate where predictions with adjusted model inputs were higher than non-bias adjusted inputs.



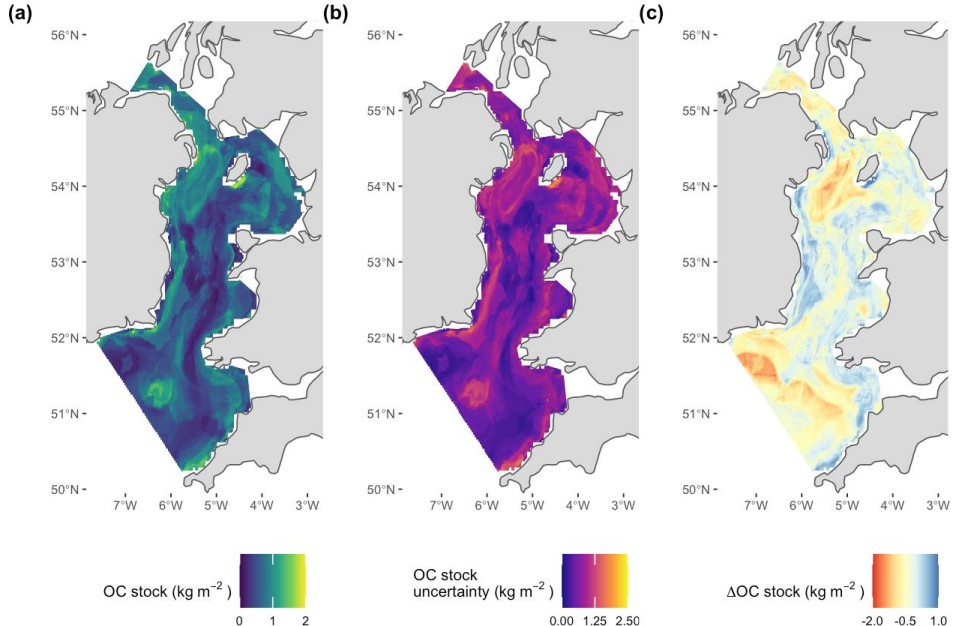

Figure 7: a) Predicted organic carbon (OC) stock using adjusted model inputs; b) the associated uncertainty and c) difference between not bias adjusted and bias adjusted predictions across the study area (difference = $OC_{stock\ pre} - OC_{stock\ post}$). Negative values indicate where predictions with adjusted model inputs were higher than non-bias adjusted inputs.