# Peer review of "Improving Marine Sediment Carbon Stock Estimates: The Role of Dry Bulk Density and Predictor Adjustments"

_EGUsphere, 2025_

## Referee Comment (RC1)

**Review of egusphere-2025-661 entitled as "Improving Marine Sediment Carbon Stock Estimates: The Role of Dry Bulk Density and Predictor Adjustments" by Chatting et al.**

This MS applied a machine learning-based DBD model to adjust DBD data and reevaluate the OC stock in the Irish Sea. They mainly compared two different ways to calculate regional OC stock. One is based on the empirical formula and unadjusted data; the other is to use a RF model in this study to generate the spatial distribution of DBD and OC stock. Their findings highlight potential overestimation of OC stock and the necessity to improve current models. Overall, this MS made a good point. However, I frequently felt it hard to follow when reading this MS, and much effort is need to make this MS easier to read. Major comments are as follows.

My most major comment is about the accuracy of the model. Although the adjusted DBD data align better with in-situ measurements, it is not convincing enough that the OC stock after adjustment is closer to the actual value. It is necessary to be clarified in different ways. Moreover, as shown in the main text, the study area lacks observational DBD data. Thus, it is questionable whether the adjusted DBD within the study area really improved the model performance and the accuracy of estimation. I also wonder the reason for just choosing the Irish Sea as the study area. It's better to emphasize these points in the main text and avoid possible vagueness.

Second, the advantages and applicability of this adjusted DBD model were not clearly illustrated. The prediction of DBD data is dependent on a lot of variables and may be difficult to scale to a larger scope. In contrast, the empirical relations are in very simple form and can be easily extended to other places. It is better to include the possibility of extending the scope of this model in the MS. Moreover, the cost and uncertainties should also be considered.

Finally, is this model robust to outliers? If not, you may need to perform data screening before model training and to see whether the performance is improved.

**Line comments**

**L15:** "over geological timescales". More studies of continental shelves are about modern environments. The focus of this MS is also in the upper 10 cm of the sediments and is not associated with geological timescales.

**L15-18:** "Shelf sediments can also be subject to…". This sentence is not related to the main content of this MS.

**L18:** Correct to "reduction".

**L19:** What data gaps? Clarify.

**L22:** "comparatively few". Compare to what? It would be better to use phrase like "sparsely" and to emphasize current data is not enough for an accurate estimation.

**L22-25:** "We compared…". This sentence is too long and can be broke up into two. You can also change the way of narrative. 1) Introduce the previous method and the shortcomings. 2) Compare your new model to the previous one.

**L25:** Clarify the depth of estimated OC stock.

**L28:** Correct to "emphasize".

**L30:** It may not be necessary to repeatedly mention "policy makers". First, lower regional OC stock is contradictory to their focus. Second, the long-term variation trend (instead of the stock) and the feedbacks to climate change are more important.

**L31:** "addressing uncertainties". This phrase is odd.

**L32:** Managing the carbon sequestration potential is not directly mentioned in the main text.

**L35-36:** You may need to cite one or two more references. For example, Hedges and Keil (1995), Burdige (2007), Bianchi et al. (2018).

**L37:** Hage et al. (2022) is about the OC burial flux in the Upper Cretaceous deltas (~75 Ma). It is far longer than the timescale of millennia.

**L42-45:** This sentence is too long and is not well joined.

**L47-51:** This sentence is too long as well. I suggest breaking it to two sentences.

**L55:** "OC per unit of s area". Typo? In addition, this sentence is too complex.

**L56:** How can DBD adjust the OC content? Maybe OC stock. Clarify.

**L67-71:** I wonder the reason for introducing the application of bias adjustments in climate models. It would be better to present the application of bias adjustments in more related fields.

**L77-78:** Temperature and salinity are not key factors in this MS.

**L87:** How to verify the OC stock estimate is improved?

**L88:** Change to "was developed by".

**L89:** dash between "un" and "adjusted". Keep consistent through the MS.

**L100:** Correct the refs.

**L104-105:** Maybe detail how to define the inshore area.

**L111:** Is 10 cm adopted in most of related studies? It would be better to consider and illustrate the corresponding timescale.

**L115:** It is better to include the number of observations (e.g., n=50). Is the conversion equation developed

exclusively from surface sediments?

**L130-132:** Again, why do you only choose the Irish Sea to test the model?

**L138-141:** It would be better to illustrate the reason of choosing these parameters. Availability or likely influence on OC stocks.

**L141:** Is the Q-Q mapping method sensitive or tolerate to outliers and extreme distribution? Clarify.

**L214-217:** This sentence is too long.

**L226:** Were the standard deviations directly summed up? Need clarification.

**L239:** It may be confusing to use both OC reservoir and OC stock. You may need to change the phrase.

**L245:** This section (4.1.1) may be inappropriate in Results part. Maybe better in Methods part.

**L289-290:** How do you estimate the uncertainty of OC stock?

**L302-304:** It would be better to point out the overestimation is within the study area. This conclusion can only be drawn after applying this adjusted DBD method to wider scope.

**Line 310:** Delete comma before "compared to" and add comma before "and".

**Line 320:** What is "OC storage dynamics"? You may change it to "OC storage variability".

**Line 328:** Add "other" before "coastal sediments".

**Line 329:** change to "sand, coarse sediments and mixed sediments".

**Line 329-331:** The interlamellar area of clay minerals may be more important for OC adsorption (e.g., Kennedy et al., 2002). Just a note.

**Line 334-345:** Refs.

**Line 346:** Space "(2024)" and "estimated".

**Line 350:** You can also take a look at the bedrock lithology in the study area. If it is dominated by sedimentary rock, significant petrogenic organic carbon associated with coarse fractions may explain the anomaly.

**Line 390:** Change to "The data collected was…".

**Line 394:** The number of observational DBD data is not enough for drawing conclusions without a doubt.

**Line 409:** There lacks a direct relation between OC storage estimation and management in the main text. Clarification is needed.

**Figure 2:** The range of color bar is too broad at present. Most OC data is within 0-2%.

**Figure 4:** Too many plots in one figure. You can consider putting some plots in the Supplement.

**Figure 5** and **Figure 6** can be combined into one figure.

**Typo**

**L55:** "OC per unit of s area" to "OC per unit of area".

**L201:** "RF's" to "RF".

**References cited**

Bianchi, T. S., Cui, X., Blair, N. E., Burdige, D. J., Eglinton, T. I., and Galy, V.: Centers of organic carbon burial and oxidation at the land-ocean interface, Org. Geochem., 115, 14–25, https://doi.org/10.1016/j.orggeochem.2017.09.008 , 2018.

Burdige, D. J.: Preservation of organic matter in marine sediments: controls, mechanisms, and an imbalance in sediment organic carbon budgets?, Chem. Rev., 107, 467–485, https://doi.org/10.1021/cr050347q , 2007.

Hage, S., Romans, B. W., Peploe, T. G. E., Poyatos-Moré, M., Haeri Ardakani, O., Bell, D., Englert, R. G., Kaempfe-Droguett, S. A., Nesbit, P. R., Sherstan, G., Synnott, D. P., and Hubbard, S. M.: High rates of organic carbon burial in submarine deltas maintained on geological timescales, Nat Geosci, 15, 919–924, https://doi.org/10.1038/s41561-022-01048-4, 2022.

Hedges, J. I. and Keil, R. G.: Sedimentary organic matter preservation: an assessment and speculative synthesis, Mar. Chem., 49, 81–115, https://doi.org/10.1016/0304-4203(95)00008-F , 1995.

Kennedy, M. J., Pevear, D. R., and Hill, R. J.: Mineral surface control of organic carbon in black shales, Science, 295, 657–660, https://doi.org/10.1126/science.1066611 , 2002.

---

## Author Comment (AC1)

We would like to thank the reviewer for their thorough and constructive feedback. Please find below our responses to RC1. Below we have provided RC1's comments (in black text) and our responses in red italic text. Where RC1 gave comments in paragraph form, we underlined key concerns in the paragraph and pasted them below with out associated response to ensure we addressed all of RC1's comments. Other than that we have addressed each one of RC1's comments in red italic text after each specific comment.

**RC1**

**Major points:**
This MS applied a machine learning-based DBD model to adjust DBD data and reevaluate the OC stock in the Irish Sea. They mainly compared two different ways to calculate regional OC stock. One is based on the empirical formula and unadjusted data; the other is to use a RF model in this study to generate the spatial distribution of DBD and OC stock. Their findings highlight potential overestimation of OC stock and the necessity to improve current models. Overall, this MS made a good point. However, I frequently felt it hard to follow when reading this MS, and much effort is need to make this MS easier to read. Major comments are as follows.

My most major comment is about the accuracy of the model. Although the adjusted DBD data align better with in-situ measurements, it is not convincing enough that the OC stock after adjustment is closer to the actual value. It is necessary to be clarified in different ways. Moreover, as shown in the main text, the study area lacks observational DBD data. Thus, it is questionable whether the adjusted DBD within the study area really improved the model performance and the accuracy of estimation. I also wonder the reason for just choosing the Irish Sea as the study area. It's better to emphasize these points in the main text and avoid possible vagueness.

Second, the advantages and applicability of this adjusted DBD model were not clearly illustrated. The prediction of DBD data is dependent on a lot of variables and may be difficult to scale to a larger scope. In contrast, the empirical relations are in very simple form and can be easily extended to other places. It is better to include the possibility of extending the scope of this model in the MS. Moreover, the cost and uncertainties should also be considered.

Finally, is this model robust to outliers? If not, you may need to perform data screening before model training and to see whether the performance is improved.

*Responses:*
*1) However, I frequently felt it hard to follow when reading this MS, and much effort is need to make this MS easier to read.*
- *The methods, results and discussion sections have been given a considerable rewrite with an emphasis on clarity for the reader. The changes made have focused on:*
  - *Shortening sentences and using clearer sentence structure*
  - *We have given the methods section a summary paragraph, which has been included at the beginning of the methods section.*

- o *The predictor abbreviations have been changed to be more intuitive and easier to remember.*
- o *Apart from these changes, does the reviewer have any specific suggestions to the structure, which would be much appreciated.*

2) *Although the adjusted DBD data align better with in-situ measurements, it is not convincing enough that the OC stock after adjustment is closer to the actual value. It is necessary to be clarified in different ways.*

- *We respectfully disagree with this comment. Organic carbon stock is not a directly measured value. It is calculated by multiplying OC content by dry bulk density (DBD) and sediment depth using the following equation:*

  *OC stock = OC content X DBD X sediment depth*

  *We have improved the inputs to this equation (OC content and DBD) as illustrated in the original manuscript and therefore the resulting OC stock estimate will be improved, too. Both our adjusted OC content and DBD models showed reduced error compared to their unadjusted counterparts. This was mentioned in the original manuscript in the results section on lines: 277 to 278 and 283 to 285.*
- *To clarify this point, we have added this text to lines 440 to 442: "In the equation for calculating OC stock (Eq. 10), DBD acts as a scaling factor that multiplies the OC content in the sediment by the amount of sediment (DBD). Therefore, it is likely that better predictions of OC content and DBD will result in more realistic estimates of OC stock."*
- *Also, we thank the reviewer for this comment and ask if the reviewer is able to provide practical advice on other ways to empirically show whether our final OC stock estimates are improvements or not.*

3) *Moreover, as shown in the main text, the study area lacks observational DBD data. Thus, it is questionable whether the adjusted DBD within the study area really improved the model performance and the accuracy of estimation.*

- *Yes, we agree with the reviewer here that a small amount of the training data, only 3% (18 of 750 data points), used to train the adjusted DBD model came from within the study area. However, we still maintain that our DBD model within the study area is more accurate than the unadjusted DBD prediction for the following reasons:*
  1. *We used data from the study area and surrounding geographic area, whereas the unadjusted version did not. The unadjusted DBD approach has frequently been used in previous modelling work (Diesing et al. 2017, 2019, Smeaton et al. 2021). However, this approach used observational data solely from the Mississippi-Alabama-Florida shelf (Jenkins 2005). Applying this relationship in the Irish Sea assumes global applicability of this relationship, which may not be the case. In contrast, our DBD RF model was developed using in situ data points from within our study area (only 3%) and the wider northwest European shelf. Additionally, in the revised manuscript we have added the following text:*
     - *On lines 326 to 330: We have carried out Area of Applicability (AOA) analysis to test whether our adjusted DBD model can be applied to our*

*study area. AOA identifies where trained machine learning models can reliably make predictions. AOA analysis calculates a dissimilarity index (DI) between prediction points and training data, which quantifies how dissimilar training data and prediction data are.*

- *On line 405 to 406: We found that 91.2% of the study area fell within the adjusted DBD model's AOA.*
- *Added on lines 472 to 474: Since >90% of the study area has predictor data comparable to training data, we can assume that the relationships 'learned' by the model during training are still applicable in the majority of the study area.*
- *We have also added AOA analysis figures in the supplementary material (Supplementary Information S5) to support these claims (figures below, which have been added to the supplementary information of the revised manuscript).*

[Figure]

- *Added on lines 462 to 465: "For example, unadjusted DBD was modelled from porosity using DBD data solely collected from the Mississippi-Alabama-Florida shelf (Jenkins 2005) and implicitly assumes global applicability of this relationship."*

2. *The unadjusted DBD model assumed a constant grain density: unadjusted DBD was modelled from porosity using the following equation:*
   *DBD = (1 - porosity) X grain density*
   *This approach involved assuming a constant grain density (2650 kg m$^{-3}$) (Diesing et al. 2017). We have added this text in the discussion section of the revised manuscript on lines 465 to 468: "Moreover, the unadjusted DBD estimate assumed a constant grain size (2650 kg m-3), however, even within similar sediment types grain density can vary, marine mud grain densities can range from 2410 to 2720 kg m$^{-3}$ (Opreanu 2003). "*

*Diesing, M., Kröger, S., Parker, R., Jenkins, C., Mason, C. and Weston, K., 2017. Predicting the standing stock of organic carbon in surface sediments of the North–West European continental shelf. Biogeochemistry, 135, pp.183-200.*

*Diesing, M., Paradis, S., Jensen, H., Thorsnes, T., Bjarnadóttir, L.R. and Knies, J., 2023. Organic Carbon Stocks and Accumulation Rates in Surface Sediments of the Norwegian Continental Margin. Authorea Preprints.*

*Jenkins, C.J., 2005, Summary of the onCALCULATION methods used in dbSEABED, in: Buczkowski, B.J., Reid, J.A., Jenkins, C.J., Reid, J.M., Williams, S.J., and Flocks, J.G., 2006, usSEABED: Gulf of Mexico and Caribbean (Puerto Rico and U.S. Virgin Islands) Offshore Surficial Sediment Data Release: U.S. Geological Survey Data Series 146, version 1.0. Online at http://pubs.usgs.gov/ds/2006/146/*

*Opreanu, G., 2003. Porosity density and other physical properties of deep-sea sediments from the Black Sea. National Institute of Marine Geology and Geo-ecology.*

*Smeaton, C., Hunt, C.A., Turrell, W.R. and Austin, W.E., 2021. Marine sedimentary carbon stocks of the United Kingdom's exclusive economic zone. Frontiers in Earth Science, 9, p.593324.*

*4) I also wonder the reason for just choosing the Irish Sea as the study area. It's better to emphasize these points in the main text and avoid possible vagueness.*
- *We acknowledge the reviewer's comment here that the reason for selecting the Irish Sea as the study area was not fully justified in the original version of the manuscript. We have added the following text to the 'Regional setting' section of the revised manuscript on lines 110 to 121:*

*"The Irish Sea was selected as the study area due to its ecological and economic importance, making it a focal point for marine resource management and conservation. It is a cross-jurisdictional region bordered by both the UK and Ireland, where overlapping policy and management frameworks further elevate its relevance for spatial planning. The Irish Sea supports some of the highest fishing intensities in Europe, with bottom otter trawling in areas such as the 'Mud Belt' and the 'Smalls' reaching an annual average of 14 hours per km² between 2009 and 2014 (ICES 2014). These same areas account for the majority of Nephrops landings in Ireland and contribute significantly to the European market, with Nephrops caught within the Irish EEZ alone valued at €53.2 million (Gerritsen and Lordan 2014). Notably, Nephrops inhabit muddy sediments, which are associated with high OC stocks (Diesing et al. 2017; Smeaton et al. 2021). Although OC stock estimates exist for the Irish Sea, they are often either coarsely resolved or geographically limited in scope (Diesing et al. 2017; Smeaton et al. 2021), highlighting the need for refined spatial modelling. This is particularly important in the Irish Sea, where a lack of data on the impacts of human activities on marine sedimentary OC stocks has been identified as a barrier to incorporating OC into marine spatial planning frameworks (Allcock et al. 2024; Crowe et al. 2023). Moreover, the Irish Sea is a data rich-region making it well suited to test and apply the spatial modelling workflow developed in this study."*

*ICES. 2014. Second Interim Report of the Working Group on Spatial Fisheries Data (WGSFD), 10–13 June 2014, ICES Headquarters, Copenhagen, Denmark. ICES CM 2014/SSGSUE:05. 102 pp. https://doi.org/10.17895/ices.pub.5683*

*Gerritsen, H.D. and Lordan, C. 2014. Atlas of Commercial Fisheries Around Ireland, Marine Institute, Ireland. ISBN 978-1-902895-56-7. 59 pp.*

**5)** _Second, the advantages and applicability of this adjusted DBD model were not clearly illustrated. The prediction of DBD data is dependent on a lot of variables and may be difficult to scale to a larger scope. It is better to include the possibility of extending the scope of this model in the MS._

- *We thank the reviewer for this concern, the main advantage of the adjusted DBD model was that the DBD RF model more closely reflects actual DBD in the Irish Sea and thus (when combined with better predictions of OC content) would provide a more realistic estimate of OC stocks than previous estimates.*

  ▪ *We have added this text on lines 483 to 493: "More reliable DBD estimates, as presented here, will result in more robust baseline assessments of marine sediment OC stocks, which are crucial to investigating the effects of human pressures on seabed OC stocks and whether managing these systems can result in meaningful emissions reductions. For example, more accurate DBD estimates can result in reducing the substantial uncertainties in $CO_2$ emissions from bottom trawling. Sala et al. (2021) and Atwood et al. (2024) both suggest that as a result of bottom trawling, significant amounts of $CO_2$ may be emitted from resuspending OC stocks in marine sediment. However, results from our study show baseline estimates of OC stocks may be substantially lower than previously reported. Additionally, impacts of trawling on marine sedimentary OC stocks has been identified as data deficient in the Irish Sea (Crowe et al. 2023), therefore in order to incorporate marine sediment OC in national marine spatial planning frameworks, more data are needed to further refine estimates to provide policy makers robust empirical evidence with which to base management decisions."*

- *The reviewer also commented that the DBD model had many inputs and would be difficult to scale to a larger scope. We agree with the reviewer on this point. This approach could mostly be applied to data-rich regions. Moreover, one of the aims of the manuscript was to use substantial amounts of legacy data to improve predictions, which the authors feel has been achieved evidenced by improved model performance metrics in both adjusted OC content and DBD models. In addition, as more data becomes available in data poor regions, RF modelling of DBD can be used to obtain more reliable estimates of marine sediment carbon stock. These points are now mentioned from lines 586 to 593 in the revised manuscript.*

**6)** _Moreover, the cost and uncertainties should also be considered._

- *We agree with the reviewer regarding the uncertainties; we did not discuss the uncertainties enough in the original manuscript. We have now added:*

- ▪ *Area of applicability (AOA) analysis, which describes the extent to which our adjusted models (OC content and DBD) should be able to perform as expected within the study area, have been added throughout the text. Text describing the rationale of AOA analysis (lines 329 to 333), the findings from it (lines 407 to 412) and its implications (lines 580 to 581) have been added throughout the revised manuscript.*
- ▪ *Text in the methods (lines 320 to 325) and discussion (lines 565 to 573) sections of the revised manuscript to highlight the uncertainties of the modelling approach. This new text aims to clarify the uncertainties considered as part of this study, and those not considered and justification for not including them and their implications for uncertainty in OC stocks.*
- ● *Regarding cost, we are not entirely sure what the reviewer means. Does the reviewer mean the economic cost of training adjusted models using legacy data? Or the economic cost of collecting more data? We would appreciate some clarity from the reviewer on this point.*

*7) Finally, is this model robust to outliers? If not, you may need to perform data screening before model training and to see whether the performance is improved.*

- ● *We performed AOA analysis for our adjusted OC content and DBD models as described between lines 329 and 333 in the revised manuscript. AOA analysis of the OC content model showed 99.5% of prediction data points' were within the AOA threshold for similarity (figures below, which have been added to the supplementary material of the manuscript). Text to clarify this has been added to lines 408 to 409 in the revised manuscript.*

[Figure]

[Figure]

- ● *In addition to AOA analysis, response data (OC content and DBD) were screened prior to being used in RF models. Below is a more detailed explanation of the data screening steps taken for response data and the corresponding lines in the revised manuscript where these inclusions have been mentioned. OC content and DBD data were:*

- *filtered to the top 10 cm of sediment profiles to account for only surficial sediment OC stocks (from line 162 to line 164)*
- *geographic locations were inspected to ensure all points fell within the study area (from line 167 to line 169)*
- *Response data were smoothed prior to training. Point data that fell within the same grid cell were averaged (Wei et al. 2022) (from line 169 to line 170)*

- *In the revised manuscript the stability of our models was tested. We have added text to describe the analysis in the methods section (lines 315 to 318): "Model stability was also tested by examining prediction consistency across repeated runs using the final selected predictors. We looked at prediction stability in the highest and lowest 15% of predicted values, we specifically chose this threshold as this is the range most susceptible to the effects of outlier." We have also added text to the results section that refers to the added analysis on model stability (lines 409 to 412): "RF model stability analysis revealed that a prediction stability of 95% was achieved with only 29 trees (the models were trained with 500 trees), indicating highly consistent predictions across runs. This low tree requirement suggests the RF models are not overly sensitive to variations in the training data"*

*Wei, Y., Qiu, X., Yazdi, M.D., Shtein, A., Shi, L., Yang, J., Peralta, A.A., Coull, B.A. and Schwartz, J.D., 2022. The impact of exposure measurement error on the estimated concentration–response relationship between long-term exposure to PM 2.5 and mortality. Environmental Health Perspectives, 130(7), p.077006.*

**Line comments**

L15: "over geological timescales". More studies of continental shelves are about modern environments. The focus of this MS is also in the upper 10 cm of the sediments and is not associated with geological timescales.
*We have removed the term 'over geological timescales'*

L15-18: "Shelf sediments can also be subject to…". This sentence is not related to the main content of this MS.
*We would like to kindly disagree with the reviewer. The aim of this sentence is to provide context and background as to why it is important to understand the OC stock in marine sediments. Even though this sentence is not related to the main content of the manuscript, it is important to provide background as to why there is a need to study marine sediment organic carbon stocks. In this context, substantial anthropogenic pressures may release the organic carbon that has been sequestered in marine sediments.*

L18: Correct to "reduction".
*This typo has been corrected.*

L19: What data gaps? Clarify.

*We have rewritten this sentence clarified in the text reasons why spatial models, specifically marine sediment organic carbon stocks, may have reduced utility. The sentence now reads: "Spatial models offer solutions to identifying organic carbon storage hotspots; however, regional predictions of OC often use global scale predictors which may have biases on smaller scales."*

L22: "comparatively few". Compare to what? It would be better to use phrase like "sparsely" and to emphasize current data is not enough for an accurate estimation.
*We agree with the reviewer's comment and have rephrased the sentence to align more with the suggestion. The edited sentence is as follows: "Moreover, estimates of dry bulk density (DBD), an important factor in calculating OC stock from sediment OC content, have large uncertainties due to a lack of in situ data for robust spatial predictions."*

L22-25: "We compared…". This sentence is too long and can be broke up into two. You can also change the way of narrative. 1) Introduce the previous method and the shortcomings. 2) Compare your new model to the previous one.
*We have rewritten this sentence to incorporate the reviewer's feedback. The sentence text now reads: "We compared the performance of two spatial models of OC stock in the Irish Sea. The first used unadjusted predictors and a commonly used empirical relationship to estimate DBD. The second spatial model incorporated bias-adjusted predictors and a machine learning DBD model trained on in situ DBD data."*

L25: Clarify the depth of estimated OC stock.
*We have now clarified in the text that we were referring to the top 10cm of sediment.*

L28: Correct to "emphasize".
*We have reworded to "highlights"*

L30: It may not be necessary to repeatedly mention "policy makers". First, lower regional OC stock is contradictory to their focus. Second, the long-term variation trend (instead of the stock) and the feedbacks to climate change are more important.
*We have removed the mention of policy makers in this instance*

L31: "addressing uncertainties". This phrase is odd.
*We have rephrased this part of the sentence to: "underscores the importance of reducing uncertainties"*

L32: Managing the carbon sequestration potential is not directly mentioned in the main text.
*We agree that carbon sequestration potential was not discussed in the text. Therefore, we have rephrased the sentence to: "…key parameters to better understand and manage OC storage potential of marine sediments."*

L35-36: You may need to cite one or two more references. For example, Hedges and Keil (1995), Burdige (2007), Bianchi et al. (2018).
*We thank the reviewer for these literature suggestions. We have added two citations to this sentence: Bianchi et al (2007) and Hedges and Keil (1995).*

L37: Hage et al. (2022) is about the OC burial flux in the Upper Cretaceous deltas (~75 Ma). It is far longer than the timescale of millennia.
*We have removed the Hage et al. (2022) citation and instead added Smeaton et al. (2021).*

L42-45: This sentence is too long and is not well joined.
*We have reworded this sentence into two sentences and improved their clarity. The text now reads: "Global estimates suggest that OC stocks in continental shelf sediment, ranging from 256 to 274 PG, are up to nine times that of coastal vegetated habitats (Atwood et al. 2020) Although still heavily debated, emissions from human pressures on marine sediments may be substantial (Hiddink et al. 2023; Sala et al. 2021)."*

L47-51: This sentence is too long as well. I suggest breaking it to two sentences.
*We have reworded this sentence as well and broken it up into two sentences. The text now reads: "Subcontinental and national scale OC stock estimates have been conducted. For example Diesing et al. (2017) reported that the Northwest European continental shelf holds between 230 and 880 Tg of OC in the top 10 cm of the sediment column, while Smeaton et al. (2021) estimated that between 456 and 592 Tg of OC were stored in surficial (0 – 10 cm) marine sediments within the United Kingdom Exclusive Economic Zone."*

L55: "OC per unit of s area". Typo? In addition, this sentence is too complex.
*We have corrected the typo in this sentence and tried to simplify the sentence.*

L56: How can DBD adjust the OC content? Maybe OC stock. Clarify.
*Yes, we thank the reviewer for noticing this mistake. It should be OC stock instead of OC content. We have now changed this mistake.*

L67-71: I wonder the reason for introducing the application of bias adjustments in climate models. It would be better to present the application of bias adjustments in more related fields.
*We thank the reviewer for this comment. We introduced the concept of bias adjustment by referencing climate modelling because this field has developed and rigorously tested these methods to correct systematic biases in large-scale model predictions, often when downscaling to regional contexts. While bias adjustments are well-established in climate science and its applied fields, for example, agricultural impact assessments, they remain underutilised in other areas of spatial environmental modelling, including marine sediment OC stock estimation. Our intention was to draw from this robust methodological framework and apply it to a novel context where global-scale predictors are similarly prone to regional biases. We have clarified this rationale in the revised text.*

L77-78: Temperature and salinity are not key factors in this MS.
*We have changed the examples given in this sentence to parameters more associated with OC stocks. The sentence now mentions sediment properties and chlorophyll-a.*

L87: How to verify the OC stock estimate is improved?
*Organic carbon (OC) stock is not directly measured but is instead calculated by multiplying OC content, dry bulk density (DBD), and sediment depth. Because of this, it is not possible*

*verify whether our adjusted OC stock estimates reflect the true values. However, we assume that improvements to the components of this calculation (OC content and DBD) would result in more accurate OC stock estimates. We have now rewritten this part of the introduction to clarify this.*

L88: Change to "was developed by".
*We have made the change suggested by the reviewer here.*

L89: dash between "un" and "adjusted". Keep consistent through the MS.
*We have removed that typo*

L100: Correct the refs.
*We have removed the mistake in the citations.*

L104-105: Maybe detail how to define the inshore area.
*As suggested by the reviewer, we have defined how the inshore area is defined in the revised manuscript.*

L111: Is 10 cm adopted in most of related studies? It would be better to consider and illustrate the corresponding timescale.
*Quantifying the top 10 cm of OC stocks has become the standard in similar larger scale OC stock estimates. We have added this clarification in the text. Additionally, we have added an approximate estimate to the length of time the top 10cm corresponds to, based on sedimentation rates in literature.*

L115: It is better to include the number of observations (e.g., n=50). Is the conversion equation developed exclusively from surface sediments?
*We have included the number of samples used in the conversion equation as well as clarifying that these samples were from surface sediments all within the top 10 cm of sediment profiles.*

L130-132: Again, why do you only choose the Irish Sea to test the model?
*We agree with the reviewer that in the original manuscript we did not provide enough justification for selecting the Irish Sea as the study area. In the revised manuscript, we have added a paragraph from lines 110 to 124 detailing why the Irish Sea was selected as the study area.*

L138-141: It would be better to illustrate the reason of choosing these parameters. Availability or likely influence on OC stocks.
*We have added text to clarify the reason for the list of predictors in section 3.1.2 "Predictor data"*

L141: Is the Q-Q mapping method sensitive or tolerate to outliers and extreme distribution? Clarify.
*We acknowledge that QQ mapping can be sensitive to extreme values, particularly when observational data are sparse or contain outliers. We have added clarification in the text to reflect this limitation and referenced Casanueva et al. (2020). To mitigate this, observational*

*data were smoothed prior to interpolation and QQ mapping to reduce the influence of extreme values. We have added this text between lines 220 and 222.*

L214-217: This sentence is too long.
*This sentence was rewritten in our general rewrite to make the manuscript as a whole easier to follow. This sentence is now split into two sentences of the revised manuscript.*

L226: Were the standard deviations directly summed up? Need clarification.
*Yes, the standard deviations were summed. We have rewritten the model uncertainty paragraph (lines 303 to 317, Section 3.7) to more clearly describe the model uncertainty process.*

L239: It may be confusing to use both OC reservoir and OC stock. You may need to change the phrase.
*We would like to keep the phrasing of OC stock and OC reservoir, both these terms have previously been used in related work. Diesing et al. (2024) used both OC stock and total reservoir when referring to the total OC stock in the study area.*

*Diesing, M., Paradis, S., Jensen, H. et al. Glacial troughs as centres of organic carbon accumulation on the Norwegian continental margin. Commun Earth Environ **5**, 327 (2024). https://doi.org/10.1038/s43247-024-01502-8.*

L245: This section (4.1.1) may be inappropriate in Results part. Maybe better in Methods part.
*As suggested by the reviewer this text has been moved to the methods section. The text that related to predictor data (OC content and DBD) was moved to "3.1.1 Response data", while the sentence that related to predictor data availability was moved to 3.2 "Bias adjusting predictors".*

L289-290: How do you estimate the uncertainty of OC stock?
*The uncertainty in OC stock was estimated by multiplying the total uncertainty in OC content with the total uncertainty in DBD and sediment depth. The following equation was used:*

*OC stock uncertainty = OC content uncertainty X DBD uncertainty X sediment depth X cell area*

*We have now clarified this in the text between lines 352 and 358.*

L302-304: It would be better to point out the overestimation is within the study area. This conclusion can only be drawn after applying this adjusted DBD method to wider scope.
*We have changed this sentence to only refer to the study area.*

Line 310: Delete comma before "compared to" and add comma before "and".
*This sentence has been rewritten and the commas have been corrected.*

Line 320: What is "OC storage dynamics"? You may change it to "OC storage variability".
*We have reworded this to align with the reviewer's suggestion.*

Line 328: Add "other" before "coastal sediments".
*We have included this suggestion*

Line 329: change to "sand, coarse sediments and mixed sediments".
*We have made this change*

Line 329-331: The interlamellar area of clay minerals may be more important for OC adsorption (e.g., Kennedy et al., 2002). Just a note.
*We thank the reviewer for this comment and agree that interlamellar surface area of certain clay minerals, particularly smectite, can be critical for OC preservation. We have modified this sentence to acknowledge the importance of interlayer surfaces.*

Line 334-345: Refs.
*We have corrected the typo with these references.*

Line 346: Space "(2024)" and "estimated".
*We have now added a space*

Line 350: You can also take a look at the bedrock lithology in the study area. If it is dominated by sedimentary rock, significant petrogenic organic carbon associated with coarse fractions may explain the anomaly.
*This comment raises the question of carbon provenance which, whilst a relevant and interesting topic, is outside the scope of this study. Generally, a number of factors need to be considered as part of a carbon provenance study. The composition of bedrock can be one of these factors. However, in relation to this study and this study area, bedrock is highly variable, and generally poorly constrained by sampling, across the Irish Sea, and is often found at significant depth (>40m) beneath a sequence of unconsolidated Quaternary sediments, which themselves can be the product of eroded glacial till. As a result, environmental factors at the seafloor and in the marine environment, including input from terrestrial sources in nearshore settings, likely play more of a role than bedrock lithology in this case.*

Line 390: Change to "The data collected was…".
*This sentence was rewritten in our rewrite and this text is no longer present in the revised manuscript.*

Line 394: The number of observational DBD data is not enough for drawing conclusions without a doubt.
*We respectfully disagree with the reviewer on this point. Our adjusted DBD model is both valid within the study area and an improvement on the widely used porosity/DBD empirical relationship. Area of Applicability (AOA) analysis (which we included in the revised manuscript) highlights that >90% of the study area is comparable to in situ data the DBD RF model was trained on. While only 3% of the data used to train the DBD RF model was within the study area, this is still an improvement compared to the widely used porosity to DBD empirical relationship that is widely used. The empirical relationship was developed using*

*data points solely from the Mississippi-Alabama-Florida shelf and assumes this relationship is globally applicable.*

Line 409: There lacks a direct relation between OC storage estimation and management in the main text. Clarification is needed.

*As suggested, we have clarified the link between our OC stock estimates and marine management. We have explicitly highlighted the relevance of improved model accuracy for marine spatial planning and policy development. The text now emphasizes that more locally robust OC stock estimates can guide seabed conservation planning and carbon vulnerability. mapping. We have added the following text to the discussion section:*

*Lines 447 to 451: "These improvements in OC stock estimation are directly relevant to marine spatial planning, particularly in the context of managing OC stocks under climate and biodiversity targets. More accurate and regionally relevant OC stock estimates can improve the reliability of national assessments, help prioritise areas for protection, and inform industry activities, such as offshore renewable energy development and fisheries management."*

*Lines 595 to 600: "Overall, our findings suggest that marine sedimentary OC stocks could be lower than previously estimated, with implications for marine spatial planning and nature-based climate solutions. Improved OC stock estimates can support more informed seabed management by identifying areas with higher carbon vulnerability or conservation potential. a conclusion with important implications for seabed management. The findings suggest that adjusting improved model inputs based on in situ data, may help refine and reduce uncertainties in model predictions to be more locally relevant."*

Figure 2: The range of color bar is too broad at present. Most OC data is within 0-2%.
*We agree with the reviewer and this figure change will be made when the fully revised manuscript is submitted*

Figure 4: Too many plots in one figure. You can consider putting some plots in the Supplement.
*We agree with the reviewer and this figure change will be made when the fully revised manuscript is submitted*

Figure 5 and Figure 6 can be combined into one figure.
*We agree with the reviewer and this figure change will be made when the fully revised manuscript is submitted*

**Typo**

L55: "OC per unit of s area" to "OC per unit of area".
*This typo has been corrected*

L201: "RF's" to "RF".
*This typo has also been corrected.*

**References cited**

Bianchi, T. S., Cui, X., Blair, N. E., Burdige, D. J., Eglinton, T. I., and Galy, V.: Centers of organic carbon burial and oxidation at the land-ocean interface, Org. Geochem., 115, 14–25, https://doi.org/10.1016/j.orggeochem.2017.09.008 , 2018.

Burdige, D. J.: Preservation of organic matter in marine sediments: controls, mechanisms, and an imbalance in sediment organic carbon budgets?, Chem. Rev., 107, 467–485, https://doi.org/10.1021/cr050347q , 2007.

Hage, S., Romans, B. W., Peploe, T. G. E., Poyatos-Moré, M., Haeri Ardakani, O., Bell, D., Englert, R. G., Kaempfe-Droguett, S. A., Nesbit, P. R., Sherstan, G., Synnott, D. P., and Hubbard, S. M.: High rates of organic carbon burial in submarine deltas maintained on geological timescales, Nat Geosci, 15, 919–924, https://doi.org/10.1038/s41561-022-01048-4, 2022.

Hedges, J. I. and Keil, R. G.: Sedimentary organic matter preservation: an assessment and speculative synthesis, Mar. Chem., 49, 81–115, https://doi.org/10.1016/0304-4203(95)00008-F , 1995.

Kennedy, M. J., Pevear, D. R., and Hill, R. J.: Mineral surface control of organic carbon in black shales, Science, 295, 657–660, https://doi.org/10.1126/science.1066611 , 2002.

---

## Author Comment (AC2)

We thank RC2 for their constructive and thorough feedback. Please find below the comments from RC2 and our responses in red italic text.

**RC2**

Review of M. Chatting et al. : "Improving Marine Sediment Carbon Stock Estimates: The Role of Dry Bulk Density and Predictor Adjustments", egusphere-2025-661

**Summary**

The present manuscript by Chatting et al. describes an improved modelling approach for calculating OC stock for marine shelf areas by upscaling point observations on OC content. Two approaches are compared: (1) The traditional one uses available globally resolved datasets for predictor variables, including dry bulk density as calculated from sediment porosity and combines it with point observations of OC content in the Irish Sea to derive the local OC stock using a random forest approach. (2) The improved one first performs a bias adjustment, which transforms global data to better represent local point observations. Dry bulk density is then extrapolated using a random forest model based on these transformed predictor data. A second random forest model is applied for the OC content and lastly dry bulk density and OC content are combined into a new OC stock estimate. The new approach shows improved performance (i.e. agreement with in situ data) in predicting dry bulk density and lowers the OC stock estimate for the region by around one third.

**General Comments**

The approach is methodologically sound. Bias adjustment has so far been mostly applied in climate models and incorporating it into marine OC stock estimation is a novel but timely application. The other modelling approach, random forest models, is well established in geospatial modelling and is an appropriate choice for the present case. The manuscript convincingly shows that bias adjustment as well as including refined DBD estimates can substantially improve our OC stock assessments.

However, the MS is in some section hard to follow, especially in the methods section. A more clear description of the modelling workflow (supported by Fig 1), a plain language summary of what bias adjustment (a central method of this study) entails, and how the success of these measures is actually measured should appear in the introduction or early in the methods sections. This way the reader can be more effectively guided through the novel data handling approach.

Another point of concern which needs to be clarified is the choice of study area. It does not become clear what the advantages of choosing the Irish Sea are, although certainly there were some. The authors mention that only 3% of DBD data (the most important predictor variable!) are available for the study area. Either expanding the scope of the modelling to the entire NW European shelf, which is the source for this DBD data, or making a good case for limiting it to the Irish Sea are needed.

Despite these points the present MS provides a useful and novel blueprint study which can help improving OC stock modelling globally, a topic with general relevance for climate science as well as biogeochemistry.

**Specific Comments**

Clarify methods: The manuscript would benefit from some more plain language step-by-step guidance throughout the methods section. E.g. in L66-72 or later, a comprehensible description of what QQ mapping entails could help familiarizing the reader with the approach.

Choice of study area: The authors should better justify their focus on the Irish Sea (e.g. in L22-23), especially as DBD seems to be sparsely available here. The reader gets the impression the entire shelf area might have been a better focus (600+ data points for DBD). Certainly there are arguments for this tighter spatial focus, which could be presented here.

Generalizability: The authors should briefly state how the findings are transferable to other geographical regions, and what it would mean for existing OC stock assessments. Is this only applicable in areas with dense available data and therefore limited to well-studied zones, or can we improve global estimates? How, e.g., would the results of Atwood et al. (2020) which are cited in the text change considering the findings of the MS? Adding a global relevance section to the discussion can help the reader better grasp the implications of the novel modelling framework.

1) A more clear description of the modelling workflow (supported by Fig 1), a plain language summary of what bias adjustment (a central method of this study) entails, and how the success of these measures is actually measured should appear in the introduction or early in the methods sections.

   o *We appreciate this comment and agree that clarity in the methods section is critical. In response, we have rewritten large sections of the manuscript for clarity, especially within the methods. This includes shortening sentences, using more intuitive terminology for predictors, and introducing a summary paragraph at the beginning of the methods section as suggested by the reviewer. We have also clarified the bias adjustment process. The simple overview we added is at the start of the methods between lines 140 and 154 in the revised manuscript and reads:*

      i. *"To estimate organic carbon (OC) stocks in surficial sediments, we developed and compared two modelling workflows. Each workflow involved predicting OC content (%) and dry bulk density (DBD), which were then combined to calculate OC stock (kg m⁻²). The key difference between the two workflows was the way environmental input data (predictors) were treated. The first approach used unadjusted, commonly available predictors and a standard DBD estimation method, while the second approach used bias-adjusted predictors, which were corrected using observational data, combined with a machine learning model to estimate DBD. A schematic overview of the workflow is provided in Fig. 1. Briefly, the process of bias-adjusting shifts the distribution of predictor data based on observational data in*

*an effort to align predictor data with in situ observations. We evaluated the success of these improvements in two ways. First, we tested whether bias-adjusted predictors more closely matched local measurements, using an error metric (Root Mean Squared Error; RMSE) which measured how far predictions deviated from in situ observations. Second, we assessed whether these improved predictors led to more accurate predictions of OC content and DBD using machine learning models, using cross-validation and RMSE. The assumption underpinning this study is that predictors that better align with in situ data would produce more reliable predictions of OC content and DBD and thus more reliable estimates of OC stock."*

- o *We have also changed many of the predictor abbreviations to be more intuitive and more easily remembered.*

2) Another point of concern which needs to be clarified is the choice of study area. It does not become clear what the advantages of choosing the Irish Sea are, although certainly there were some. The authors mention that only 3% of DBD data (the most important predictor variable!) are available for the study area. Either expanding the scope of the modelling to the entire NW European shelf, which is the source for this DBD data, or making a good case for limiting it to the Irish Sea are needed.

- o *Thank you for this observation. We have expanded our justification for choosing the Irish Sea in the regional setting section (lines 110–124), citing its ecological and economic importance, heavy bottom-trawling pressure, and known high-OC mud sediments.*
- o *Also we acknowledge that only 3% of the DBD training data come from within the Irish Sea. However, we argue that our adjusted DBD model is an improvement over the unadjusted method for several reasons:*
  - i. *The unadjusted method, which has frequently been used in previous work (Diesing et al. 2017, Diesing et al. 2021, Smeaton et al. 2021), uses data solely from the Mississippi-Alabama-Florida shelf, assuming global applicability, which may not be valid for the Irish Sea.*
  - ii. *Our adjusted DBD model includes data from the Irish Sea and surrounding NW European Shelf and underwent Area of Applicability (AOA) analysis, showing that 91.2% of the study area is within model applicability.*
  - iii. *We now include these AOA figures (Supplementary S5, figures below) and corresponding clarifying text in the manuscript.*

[Figure]

[Figure]

*Diesing, M., Kröger, S., Parker, R., Jenkins, C., Mason, C. and Weston, K., 2017. Predicting the standing stock of organic carbon in surface sediments of the North–West European continental shelf. Biogeochemistry, 135, pp.183-200.*

*Diesing, M., Paradis, S., Jensen, H., Thorsnes, T., Bjarnadóttir, L.R. and Knies, J., 2023. Organic Carbon Stocks and Accumulation Rates in Surface Sediments of the Norwegian Continental Margin. Authorea Preprints.*

*Smeaton, C., Hunt, C.A., Turrell, W.R. and Austin, W.E., 2021. Marine sedimentary carbon stocks of the United Kingdom's exclusive economic zone. Frontiers in Earth Science, 9, p.593324.*

3) The authors should briefly state how the findings are transferable to other geographical regions, and what it would mean for existing OC stock assessments. Is this only applicable in areas with dense available data and therefore limited to well-studied zones, or can we improve global estimates? Is this only applicable in areas with dense available data and therefore limited to well-studied zones, or can we improve global estimates? How, e.g., would the results of Atwood et al. (2020) which are cited in the text change considering the findings of the MS? Adding a global relevance section to the discussion can help the reader better grasp the implications of the novel modelling framework.

   o *We agree and have included text in the discussion (lines 586–593) that our method relies on substantial data inputs to improve OC stock estimates. The improved methodology is applicable where large amounts of in situ data are available. We note that as more data becomes available in data-poor regions this method may be applied. Additionally, we have also included (lines 445–447) text to state that previous work using the porosity empirical relationship may have represented overestimates of DBD and consequently OC stock.*

**Line comments**

L15:    the study addresses the short-term C cycle, not the geological one, consider rephrasing
*We have removed the term 'over geological timescales'*

L19:    "Data gaps" may be misleading here. The study does not collect new data but rather improves the interpretation of existing data, rephrase
*We have rephrased this sentence which now reads: "… however, regional predictions of OC often use global scale predictors which may have biases on smaller scales, reducing their utility for practical management decisions."*

L22-23: State why Irish Sea was chosen; maybe due to sufficient data availability?
*We disagree with the reviewer's intent here. We do not think that justification for the Irish Sea should be given in the abstract. We have removed "in the Irish Sea" from the sentence. Instead we have provided a more thorough justification for selecting the Irish Sea as the study is in section 2: Regional setting.*

L28:    "emphasize" instead of "emphasizes" as it refers to "findings"
*This error has been changed. We have also changed the word to "highlight" instead of "emphasize".*

L29:    ensure consistent formatting of in situ (throughout the MS)
*We have gone through the manuscript and standardized the formatting of "in situ"*

L30:    consider removing "for policy makers", as many more stakeholders are interested in improved OC stock assessment
*"for policy makers" has been removed from the text*

L49-50: harmonize phrasing "uppermost 10 cm", "surficial 0-10 cm" and "top 0.1 m" (in methods section) all refer to the same and should be consistent
*We have standardized all mentions to the "top 10 cm", apart from one occurrence. We have kept "surficial (0 - 10 cm)" as it appears just after a sentence with "top 10 cm", we wanted to avoid repetition so have kept "surficial (0 - 10 cm)" for this one instance.*

L56:    possibly use "OC stock" instead of "OC content"; the sentence is hard to read
*We have rewritten this sentence to be clearer for the reader. It now reads: "DBD is a scaling factor on OC content and adjusted the OC stock in a given volume based on the density of sediment."*
L58:    "sediment density" should be "soil density"
*We have rephrased here to "soil density"*

L59:    consider removing "however" to improve flow
*We have removed "however" from the sentence*

L66-72: The repeated mention of "climate models" and references about them can be drastically shortened. Instead it could include a one sentence, plain-language summary of what bias adjustment entails

*We have simplified the text in this paragraph. It now reads: "To address these discrepancies, bias adjustment techniques are commonly used in other scientific disciplines, for example in climate science, where large-scale models are adjusted to better align with local observational data (Laux et al., 2021; Luo et al., 2018). Bias adjustments reduce systematic errors in model outputs and ensures that projections match local conditions and are reliable for practical applications (Laux et al., 2021). Bias adjustments have been used to improve climate model utility in agricultural impact assessments, such as predicting planting dates and crop suitability in water-limited regions; to correct overestimations in soil moisture models and to improve predictions in sea ice thickness (Laux et al., 2021; Lee and Im, 2015; Mu et al., 2018). Despite their widespread use in climate science, bias adjustment methods are underutilised in other areas of spatial environmental modelling, including OC stock modelling." Additionally, we have provided a brief plain language of bias adjustment at the start of the methods section (lines 146 to 148)*

L78:    If OC and mud content measurements also use similar instrumentation maybe mention this (instead)
*We have rephrased to sediment properties and OC content*

L87:    Be more clear how model improvement is assessed
*We have added text at the start of the methods section (lines 148 to 152) that gives a plain language summary of how we assessed potential improvement in environmental predictors and random forest models. The text reads: "We evaluated the success of these improvements in two ways. First, we tested whether bias-adjusted predictors more closely matched local measurements, using an error metric (Root Mean Squared Error; RMSE) which measured how far predictions deviated from in situ observations. Second, we assessed whether these improved predictors led to more accurate predictions of OC content and DBD using machine learning models, using cross-validation and RMSE. The assumption underpinning this study is that predictors that better align with in situ data would produce more reliable predictions of OC content and DBD and thus more reliable estimates of OC stock."*

L97:    Does this mosaic of sediment types help modelling here?
*Having variability in the sediment types, so long as that varies with parameters of interest (OC content and DBD) does help spatial modelling here. With little variation in sediment types but still substantial variation in OC content and DBD, it would be more difficult to use sediment type as a predictor of OC content and DBD. It is likely other parameters would be more useful. This makes intuitive sense, as in this hypothetical situation, sediment type would not be causing heterogeneity in OC content and DBD.*

L104-105: Briefly mention how inshore area is defined in this data
*We have added clarification on how the inshore area was defined on line 137 of the revised manuscript.*

L147: Add a sentence as plain-language summary of QQ mapping
*We have edited the sentence here to make it easier for the reader to follow on lines 213 to 215 on the revised manuscript: "This approach aligns the quantiles in observational and modelled data and preserves the spatial patterns of the original data while aligning their*

*statistical distribution with in situ data, and has QQ mapping bias adjusted models have been shown to outperform un-adjusted models (Ngai et al., 2017)."*

L153-165: Consider moving to Supplementary, log ratio transformation is a standard procedure in compositional data and not crucial to the presented modelling approach

L192: Is this the mud content from spatial averaging? Clarify
*We have clarified here that we were referring to spatially averaged mud.*

L206: refer to Fig 4 here (may be Fig 3 then)
*We disagree with the reviewer on this point. As this is the methods section and still describing the concept of important predictor selection we do not think it is appropriate to refer to the plot that displays the important predictors partial dependence on OC content.*

L214: could this k fold CV be replaced with the NNDM LOO CV, which is introduced later and said to perform better?
*Here we are referring to the random k fold CV and that it does not perform as well as NNDM LOO CV. To make it clearer for the reader, we have rephrased the text to read (lines 299 to 303): "Random k fold cross-validation, can produce overly optimistic performance estimates by splitting spatially autocorrelated data across training and testing sets. By contrast, NNDM ensures spatial independence between folds, providing better estimates of model performance on spatially independent data (Milà et al., 2022)."*

L236: "by a grid cell" instead of "by grid cell"
*This typo has been corrected*

L241: be more explicit than "all possible combinations", there are 4 combinations; OC adj/unadj with DBD adj/unadj correct?
*We have clarified in the text that we are talking about 4 possible combinations of adjusted vs unadjusted OC and DBD models.*

L273: Sort plots by predictor importance in Figure 4
*We agree with the reviewer and will sort the plots by predictor importance in the revised manuscript*

L277: How would this approach perform in regions with even fewer DBD observations?
*We have clarified in the discussion section (lines 582 to 589) the limitations of having fewer DBD points, which is why AOA analysis was performed. The text reads: "The refined estimates presented in this study rely on large amounts of in situ data and environmental predictors, making this approach most suitable for data-rich regions. Within our study area, the limited availability of DBD measurements required the use of an Area of Applicability (AOA) analysis to assess whether the adjusted DBD model could be reliably applied—highlighting potential limitations of this approach in data-poor settings. Nonetheless, our findings demonstrate that where sufficient observational data are available, OC stock estimates can be substantially improved. As more in situ datasets are generated in currently*

*under-sampled regions, this modelling framework can be replicated and further refined to support better-informed carbon assessments."*

L318-320: Clarify how the presented findings would influence their estimates
*We have included text between lines 441 and 443 that suggests how these findings would impact other estimates. "Additionally, these findings highlight the importance of using improved DBD models and suggests that previous estimates of OC stock that used the porosity empirical relationship may represent overestimates."*

L326: move the definition of mud to its first mention in the MS
*We have moved the definition to earlier in the manuscript. It is now on line 221 of the revised manuscript.*

L331-335: The sentence is long and unclear; what is the "topography" of a mineral grain? Also the references need sorting
*The sentence has been rephrased. It now reads: "The capacity for sediments to bind OC through clay-OC interactions can also vary with different mineral phases occurring in sediments, varying in particle-size as well as surface area, charge and distribution, and subsequent geochemical conditions constraining these characteristics (e.g. pH and ionic strength of pore water)."*

*Also the references have been corrected.*

L346: Space missing between "(2024)" and "estimated"
*We have corrected this typo*

L356: "resuspension" instead of "suspension"
*We have rewritten this sentence and it no longer needs correction*

L362-63: "needs" instead of "need"
*We have rewritten the sentence to be clearer.*

L388: "carries" instead of "carry"
*We do not agree with the reviewer here. The word "carry" here refers to the "predictions" mentioned at the start of the sentence. Since "predictions" is plural, "carry" should be used, not "carries"*

L398: rephrase "increased in situ data" to "increased availability of in situ data"
*We have edited this sentence. It now reads: "As more in situ datasets become available in currently under-sampled regions, this modelling framework can be replicated and further refined to support better-informed carbon assessments."*

Figure 2: It seems the yellow shade is not reached/used, maybe adjust the colormap. Also the thick outline is not very visually appealing and might obscure data points. A dashed line should be tested.
*We agree with the reviewer and this figure change will be made when the fully revised manuscript is submitted*

Figure 4: The plot is not very visually appealing, it is not clear what the letters refer to. Maybe the distance between the a and b row can be slightly increases. Plots should be sorted in the order of relative parameter importance. Also the figure caption for 4 c is not correctly describing the presented plots.

*We agree with the reviewer and this figure change will be made when the fully revised manuscript is submitted*

Figure 5, 6 and 7 all show the same region, but different parameters. Maybe combining them all into one large, page filling figure would allow the reader to better appreciate all present trends at once. Another side note is, that the text mentions the "Isle of Man", which is not labelled in a map and might be unfamiliar to many readers.

*We agree with the reviewer and this figure change will be made when the fully revised manuscript is submitted*

---

## Author Response (AR2)

We would like to thank the reviewers for their continued constructive feedback. Please find below our responses to Reviewer 1. Below we have provided Reviewer 1's comments (in black text) and our responses in red italic text.

**Reviewer 1**

Only a few technical corrections remain. All line numbers refer to the authors' tracked changes document:

L88 - Add "and" between "sediment properties" and "OC content" This typo has been corrected

L99 - Typo: "these" instead of "tehese" This typo has been corrected

L113 - Typo: "water" instead of "otter"

We respectfully disagree with the reviewer here. This is not a typo, "bottom otter trawling" is a type of fishing gear that is widely used in the Irish Sea. We have now clarified this in the text

L120/121 - Consider rephrasing "lack of data" as L123 refers to the area as "data rich" *This sentence has now been re-phrased so as not to sound contradictory. It now reads:*

"This is particularly important in the Irish Sea, where although the region is generally datarich, limited information on the impacts of human activities on marine sedimentary OC stocks has been identified as a barrier to incorporating OC into marine spatial planning frameworks (Allcock et al., 2024; Crowe et al., 2023). Moreover, the availability of broader environmental datasets makes the Irish Sea well suited to test and apply the spatial modelling workflow developed in this study."

L306 - Typo? NNDM is performing well on spatially "dependent", locally autocorrelated data, not "independent"

We respectfully disagree with the reviewer here. This is not a typo. The idea of NNDM is to achieve more realistic estimates of model performance than random cross validation. During NNDM the training/testing folds are split to make sure they are spatially independent. So the model is tested on data that is spatially uncorrelated to training data, which gives a more realistic estimate of model performance. Conversely random cross validation creates training/testing splits that likely contain spatially correlated data in both training and testing datasets. Thus the model is trained and tested on similar data, which artificially inflates model performance metrics and does not give a fair reflection of how the model performs in unknown areas. We have now clarified this in the text.

L318 - Provide a reference for the statement on the "15% interval" We have now added a relevant citation for this statement.

L400 - Typo: "MSE" instead of "RMSE", as this metric was introduced shortly before and applied to the other cases

**This typo has been corrected.**

L431 - The "Mudbelt" and "Smalls" areas are referred to slightly differently here compared to L423 and L572; please harmonize

References to the western Irish Sea 'mud belt' and the 'Smalls' have been harmonized throughout the manuscript.

L473 - Typo: grain "density" not "size" *This typo has been corrected.*

L543 - Add "that" or "which" before "impact" *This mistake has been corrected.*

L555 - Do not delete "to"

This mistake has been corrected.

We thank RC2 for their continued constructive feedback. Please find below the comments from Reviewer 2 and our responses in red italic text.

**Reviewer 2**

The first round of revisions have substantially improved the quality of the MS. I would support publication after a few minor revisions.

1. Abstract, L18-22. First, the flow of these sentences can be improved. For example "Moreover" is not accurate, maybe another word here. Second, DBD estimates are mostly based on global data but used in regional OC stock estimates. This point should be clearly stated. So these sentences need to be rephrased.

We have rephrased these sentences to be clearer and more explicit with regards to previous DBD estimates. The text now reads:

"Spatial models offer solutions to identifying organic carbon storage hotspots; however, regional predictions of OC often rely on global scale predictors which may have biases on smaller scales, reducing their utility for practical management decisions. In addition, estimates of dry bulk density (DBD), an important factor in calculating OC stock from sediment OC content, are typically derived from an empirical relationship developed in one region and applied elsewhere, rather than from local in situ data, leading considerable uncertainty in regional OC stock estimates."

- 2. Introduction. Some expressions are not clear.
  - a. L78 sediment properties OC content.

This was a mistake. We have added the word "and" between sediment properties and OC content.

b. L88 tehse inputs.

This was a typo. We have now corrected this. It should have read "these inputs"

3. Conclusion. This section is now like implications. Important results and data (about DBD estimates) should be emphasized again in this section.

We have edited the conclusion section of the manuscript to now include important results of the study, particularly with regards to DBD estiamtes. This section now reads:

"Overall, our findings suggest that marine sedimentary OC stocks could be lower than previously estimated, with implications for marine spatial planning and nature-based climate solutions. A key result of this study is that uncertainties in dry bulk density (DBD) estimates strongly influence OC stock predictions. We show that reliance on previously developed empirical relationships for DBD can introduce substantial error, underscoring the need for regionally relevant data. Improved OC stock estimates, grounded in more accurate DBD values, can support more informed seabed management by identifying areas with higher carbon vulnerability or conservation potential. a conclusion with important implications for seabed management. The findings suggest that adjusting improving model inputs based on in situ data, may help refine reduce uncertainties in model predictions to be more locally relevant. We highlight the critical role that accurate DBD estimates play in determining OC stock. Moving forward, more comprehensive in situ DBD measurements and

refined DBD models are essential for improving the accuracy of OC stock predictions. Alternatively, OC stocks could be calculated directly per sediment core, reducing the number of models needed to estimate OC stocks, thus reducing uncertainty in final estimates. These efforts will be instrumental in developing better strategies for managing marine sedimentary OC stocks."